# Synthetic analysis of trophic diversity and evolution in Enantiornithes with new insights from Bohaiornithidae

Case Vincent Miller[1]*, Jen A Bright[2], Xiaoli Wang[3,4], Xiaoting Zheng[3,4], Michael Pittman[5]*

[1]Department of Earth Sciences, The University of Hong Kong, Hong Kong SAR, China; [2]School of Natural Sciences, University of Hull, Hull, United Kingdom; [3]Institute of Geology and Paleontology, Linyi University, Linyi, China; [4]Shandong Tianyu Museum of Nature, Shandong, China; [5]School of Life Sciences, The Chinese University of Hong Kong, Hong Kong SAR, China

*For correspondence:
Case.Miller@connect.hku.hk
(CVincentM);
mpittman@cuhk.edu.hk (MP)

Competing interest: The authors declare that no competing interests exist.

**Abstract** Enantiornithines were the dominant birds of the Mesozoic, but understanding of their diet is still tenuous. We introduce new data on the enantiornithine family Bohaiornithidae, famous for their large size and powerfully built teeth and claws. In tandem with previously published data, we comment on the breadth of enantiornithine ecology and potential patterns in which it evolved. Body mass, jaw mechanical advantage, finite element analysis of the jaw, and traditional morphometrics of the claws and skull are compared between bohaiornithids and living birds. We find bohaiornithids to be more ecologically diverse than any other enantiornithine family: *Bohaiornis* and *Parabohaiornis* are similar to living plant-eating birds; *Longusunguis* resembles raptorial carnivores; *Zhouornis* is similar to both fruit-eating birds and generalist feeders; and *Shenqiornis* and *Sulcavis* plausibly ate fish, plants, or a mix of both. We predict the ancestral enantiornithine bird to have been a generalist which ate a wide variety of foods. However, more quantitative data from across the enantiornithine tree is needed to refine this prediction. By the Early Cretaceous, enantiornithine birds had diversified into a variety of ecological niches like crown birds after the K-Pg extinction, adding to the evidence that traits unique to crown birds cannot completely explain their ecological success.

## eLife assessment

This **important** study explores numerous lines of evidence for the surprisingly diverse diets of a group of toothed birds that lived over 100 million years ago. The large amount of data the authors collected forms a **solid** dataset. The methods might in principle be extensible to other limbed vertebrates, although there are concerns regarding some of the details. The article will be of interest to colleagues studying ecological evolution in birds or dinosaurs more generally, as well as to anyone studying the impact of the mass extinction event 66 million years ago.

## Introduction

The diet of crown birds is extremely broad, and dietary evolution within the crown is a complex mosaic (*Felice et al., 2019*). However, dietary evolution in birds outside the crown is poorly understood, largely because the diet of most Mesozoic birds remains speculative (*Miller and Pittman, 2021*). Some recent studies have begun to elucidate this matter (*O'Connor, 2019a*; *O'Connor and Zhou, 2020c*; *Miller et al., 2020*; *Hu et al., 2022*; *Marugán-Lobón and Chiappe, 2022*). The most progress

**eLife digest** The birds living in the world today are only a small part of the larger bird family tree. Around 120 to 65 million years ago, when dinosaurs and other large reptiles roamed the world, the ancestors of modern-day birds were actually rather rare. Instead, another now extinct group of birds called the Enantiornithes (meaning "opposite birds") were the most common birds.

Many researchers believe that Enantiornithes may have filled similar roles in ancient ecosystems as living birds do today. For example, some may have hunted other birds or animals, while some may have eaten only plants. Some may have specialized at eating a few specific foods while others may have been 'generalists' that ate many different foods. However, some of the physical features of Enantiornithes set them apart from modern-day birds. For example, unlike living birds, Enantiornithes had teeth and their wings were also constructed very differently.

Previous studies suggest that one group of these extinct birds most likely ate insects and another group most likely ate fish, but it remains unclear what variety of foods opposite birds as a whole may have consumed. Miller et al. compared the jaws, claws and various other physical features of fossils from six additional species of opposite birds with the skeletons of modern birds to infer what the diets of these opposite birds may have been.

This approach revealed that Enantiornithes may have had a wide variety of different diets. The researchers found that two species probably ate plants, another species most likely ate meat, and another one likely ate a mixture of both. With a large sample across Enantiornithes, Miller et al. were able to predict the diet of their common ancestor. They found the common ancestor to most likely be a 'generalist' eating variety of foods and that some species subsequently evolved to have more specialist diets.

Opposite birds probably played many different roles in ecosystems, like living birds do today. Therefore, a better understanding how Enantiornithes evolved may shed light on the factors that have influenced the evolution of modern-day birds. This may aid future conservation efforts to target birds whose descendants may be able to take up the ecological roles of other species that go extinct.

has been made among enantiornithines (*Miller et al., 2022*; *Miller et al., 2023*; *Clark et al., 2023*), the most abundant and speciose birds in the Cretaceous (*Pittman, 2020a*). This progress only amounts to an examination of 12 of the over 100 described enantiornithine species (*Pittman, 2020a*), though. This has limited any large-scale examinations of the overall trophic diversity of enantiornithines and the patterns in which they diversified. Ideally, the next step to answering these questions is to examine a large group of enantiornithines which are phylogenetically intermediate to the previously studied families. The enantiornithine family Bohaiornithidae (*Figure 1*, centre) fits both of these requirements, and thus serves as an ideal stepping stone to a large-scale understanding of enantiornithine ecology.

Bohaiornithids are iconic among enantiornithines for their robust teeth, large claws, large size (*Wang, 2014a*), and iridescent colouration (*Peteya et al., 2017*). Their robust teeth have led the clade to be interpreted as durophagous (*O'Connor and Chiappe, 2011a*; *O'Connor et al., 2013*; *Chiappe and Meng, 2016*; *Zhou et al., 2021*), while their large claws and large body size lead to raptorial interpretations (*Wang, 2014a*; *Chiappe and Meng, 2016*; *Li et al., 2014*). Bohaiornithidae has a troubled taxonomic history, with authors proposing the clade actually represents an evolutionary grade (*Chiappe et al., 2019*; *Wang et al., 2022b*) or at least a group in need of phylogenetic redefinition (*Liu et al., 2022*). But across the literature, the original six taxa referred to Bohaiornithidae (*Bohaiornis*, *Longusunguis*, *Parabohaiornis*, *Shenqiornis*, *Sulcavis*, and *Zhouornis)* (*Wang, 2014a*; *Figure 2*) have consistently resolved as closely related taxa (*Table 1*), so for the purposes of this work we will refer to this group as a clade (see 'Methods' for further justification). With the recently described *Beiguornis* (*Wang, 2022a*), Bohaiornithidae is the most speciose family of enantiornithine birds. Given their unusual morphology and apparent ecological success, Bohaiornithidae presents an ideal study topic for Mesozoic bird diet.

To investigate bohaiornithid diet, we utilise four quantitative diet proxies: body mass, mechanical advantage (MA) and related functional indices of the jaws, finite element analysis (FEA) modelling the jaws during a bite, and traditional morphometric (TM) analysis of claw shape and size. Size has a strong effect on birds' diets (*Navalón et al., 2019*; *Pigot et al., 2020*; *Natale and Slater, 2022*), so

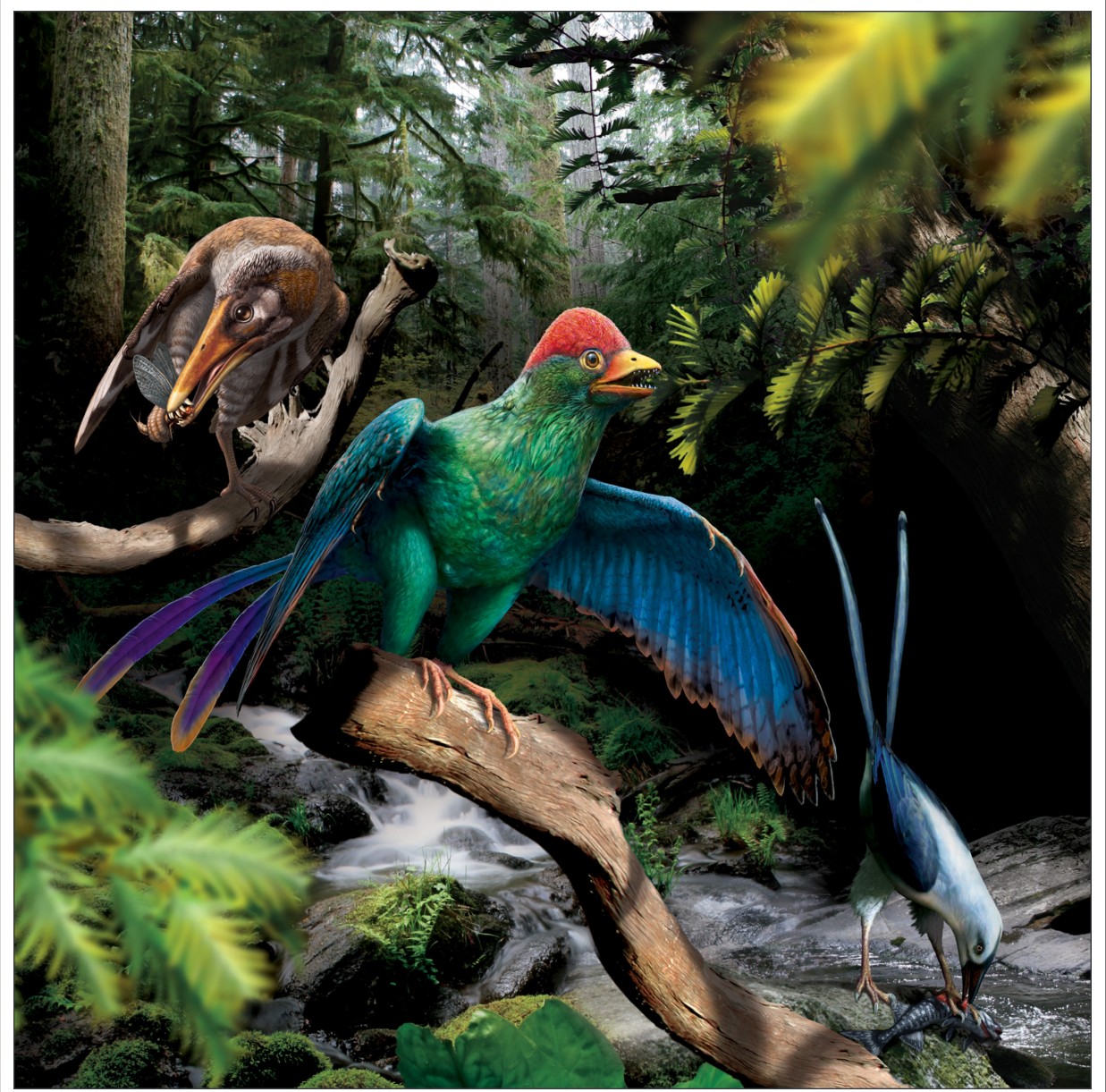

**Figure 1.** Life reconstruction of enantiornithine birds feeding. *Longipteryx* (left), *Bohaiornis* (centre), and *Pengornis* (right) are pictured in the Early Cretaceous forests of northeastern China, roughly 120 million years ago. *Bohaiornis* is depicted feeding on cypress (Cupressaceae, **Ding et al., 2016**) leaves after the findings in this work. *Longipteryx* is depicted feeding on the mayfly *Epicharmeropsis hexavenulosus* (**Huang et al., 2007**) after (**Miller et al., 2022**). *Pengornis* is depicted feeding on the fish *Lycoptera davidi* (**Chang and Miao, 2004**) after **Miller et al., 2023**.

estimating the mass of extinct birds (**Serrano et al., 2015**) helps narrow dietary possibilities. Functional indices are ratios of measures of an animal that inform the mechanical efficiency of body parts to exert or withstand certain forces. Most commonly, the functional index used in ecology is MA of the jaw, looking at trade-offs between bite speed and force (**Stayton, 2006**; **Corbin et al., 2015**; **Adams et al., 2019**). Herein three versions of MA are collected alongside three other functional indices which have previously discriminated animal diet (**Miller and Pittman, 2021**; **Ma, 2020**). FEA is an engineering tool used to model forces acting on irregular structures (**Bathe, 2014**). Here it is used to model bird jaws during a bite. If a model experiences less strain under the same relative load as another model, its shape can withstand a greater force before failure. By maintaining a constant relative load for FEA models, models can be compared in terms of relative strength (**Dumont et al., 2009**; **Bright, 2014**). TM describes the shape of animal parts with measurements relevant to their ecological

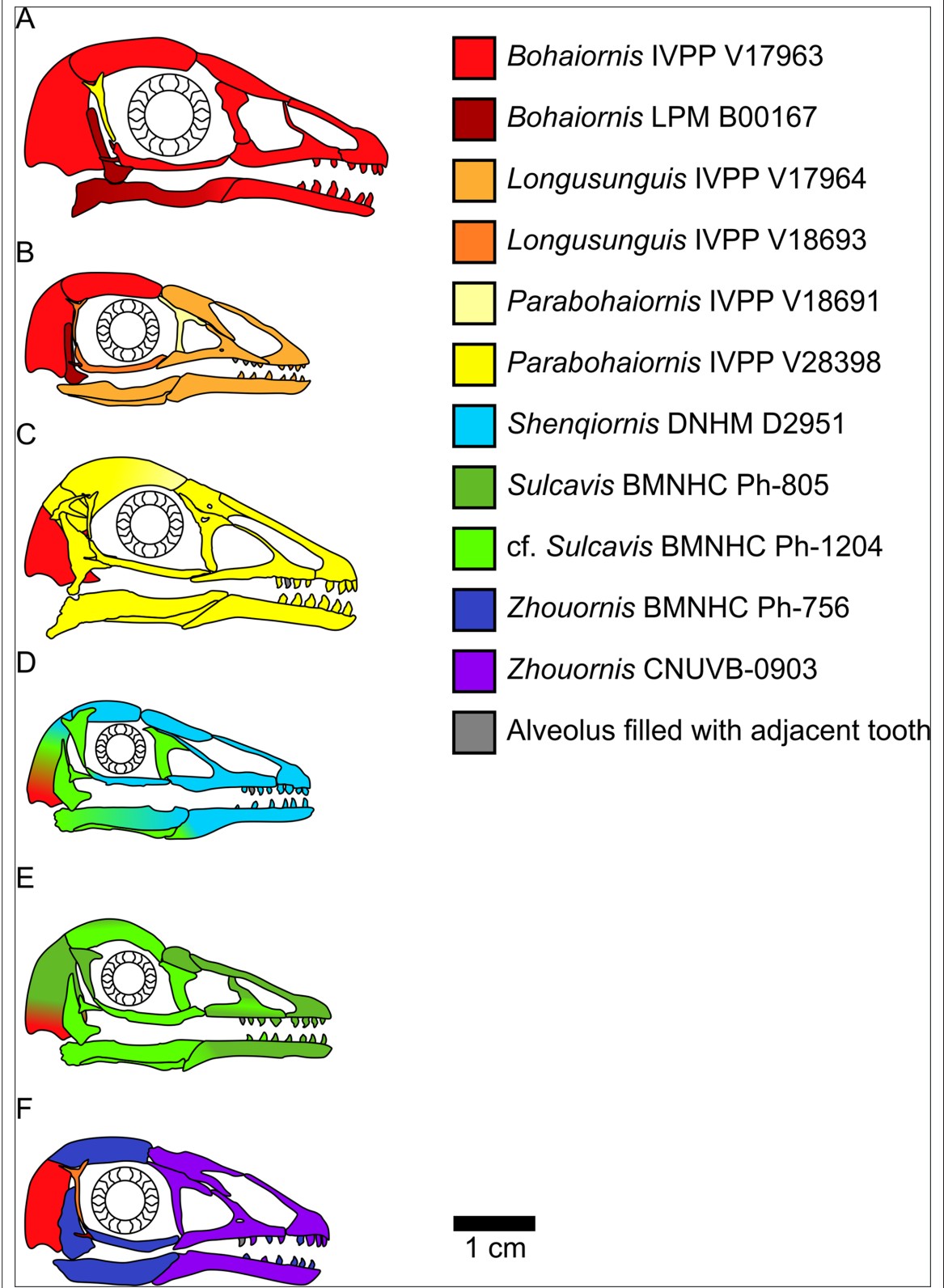

**Figure 2.** Bohaiornithid skull reconstructions used for mechanical advantage (MA) and finite element analysis (FEA) calculations in this study. Reconstructions are of *Bohaiornis* (**A**), *Longusunguis* (**B**), *Parabohaiornis* (**C**), *Shenqiornis* (**D**), *Sulcavis* (**E**), and *Zhouornis* (**F**). Different colours indicate elements taken from different individual specimens. All sclerotic rings are based on *Longipteryx* specimen BMNHC Ph-930B. See the 'Methods' section for more details on reconstruction. Scale for each reconstruction is based on the individual which makes up the largest portion of the reconstruction.

*Figure 2 continued on next page*

*Figure 2 continued*

The online version of this article includes the following figure supplement(s) for figure 2:

**Figure supplement 1.** Reconstructions of bohaiornithid skulls for sensitivity analysis of the quadrate's position, which is uncertain due to disarticulation in all specimens.

role (*Marcus, 1990*). Here, measurements are taken of the curvature and relative size of pedal unguals which can differentiate between non-raptorial and raptorial birds (*Csermely and Rossi, 2006*; *Fowler et al., 2009*; *Csermely et al., 2012*; *Cobb and Sellers, 2020*) and, potentially, between specific types of raptorial behaviour (*Miller et al., 2022*; *Fowler et al., 2009*; *Einoder and Richardson, 2007*).

We expand the framework used in our past works (*Miller et al., 2022*; *Miller et al., 2023*) by including traditional morphometrics of the skull. This line of evidence was recently used successfully in reconstructing the diet of longipterygid enantiornithines (*Clark et al., 2023*), so we expect it to be useful here as well. We incorporate the extant data from *Clark et al., 2023*, collect new data for bohaiornithids, and analyse the data with a modified version of their methodology. Notably, we use alternate sources of body mass and diet data and adjust the data in an attempt to remove the effects of body size.

These proxies are of little use without reference values, so comparative data is taken from nearly 200 extant birds (*Miller et al., 2022*; *Miller et al., 2023*), including tinamous, flamingos, turacos, strisores, and songbirds, among others. These birds are also ecologically diverse, with diet categories based on the EltonTraits 1.0 database of bird diet (*Wilman et al., 2014*). *Table 2* provides cut-offs for diet assignment. Claw shape is not expected to correlate with diet but the use of talons. We follow the classification of *Miller et al., 2023* separating raptorial birds into raptors taking small prey (which can be completely encircled in the pes) and those hunting large prey (which cannot be encircled by the pes) based on feeding records in the Birds of the World database (*Billerman, 2023*). With the recent publication of the bird database AVONET (*Tobias et al., 2022*), we are able to expand our dataset comparing mass and diet to 8758 of the 9994 recognised species of birds.

After each proxy is analysed, they are synthesised into a set of likely diets agreed upon by different evidence. Combining multiple lines of evidence allows for more precise and confident diet assignments than any single line can provide. Using this framework (*Miller and Pittman, 2021*) we quantitatively test hypotheses of durophagy and raptorial behaviour in bohaiornithids.

Once dietary predictions are made for Bohaiornithidae, we can begin to examine large-scale trends in enantiornithine ecology via ancestral state reconstruction. Each of the diet proxies above can be predicted for the common ancestor of Enantiornithes and placed into the same framework as any individual species, to create a diet hypothesis for the common ancestor. However, one would reasonably question if this sample size is large enough to intuit an ancestral diet reconstruction. To better visualise the uncertainty of our relatively small sample size, we also include a larger tree with qualitative diet assignments. This tree assumes published diet hypotheses are all correct as a test of what sampling density is necessary to produce confident ancestral state reconstructions of diet.

## Results

### Body mass

Of the 9994 bird species in *Tobias et al., 2022*, 1236 birds either had no mass data or did not fall into one of our diet categories (*Table 2*), leaving a sample size of 8758. Bird masses separated into four significantly different combinations of diet categories using phylogenetic honest significant differences (HSD) (*Miller et al., 2022*; *Collyer and Adams, 2018*): (a) nectarivores; (b) granivores and invertivores; (c) frugivores and generalists; and (d) folivores and tetrapod hunters. Piscivores and scavengers were not significantly different from groups b or c. Optimising the Youden index (*Fluss et al., 2005*), cut-off points between these diet groups are as follows: between a and b, 9 g (95% confidence interval [CI] 8–10 g); between b and c, 56 g (95% CI 41–63 g); and between c and d, 265 g (95% CI 162–303 g). Mass of birds by diet and cut-off points between them are visualised in *Figure 3*. Significant phylogenetic signal was found in extant bird mass, with trends resembling those expected under a Brownian motion model of evolution (K = 0.93; *Supplementary file 1*). Estimated bohaiornithid body masses are provided in *Table 3*. Masses range from 91 g to 905 g, x̄ = 287 g, though the largest

**Table 1.** Summary of taxa included in Bohaiornithidae.

'Bohaiornithidae' is used in an informal term to refer to any distinct clade containing *Bohaiornis*, as the strict clade definition is unstable (*Liu et al., 2022*). If multiple members of the other five originally defined bohaiornithids (*Wang et al., 2014b*) resolved near *Bohaiornis*, 'Bohaiornithidae' was considered to be the least inclusive clade containing all of them with a tolerance of two bird taxa not in the original six between any two internal nodes.

| Status | |
| --- | --- |
| 'Bohaiornithidae' | + |
| Sister to 'Bohaiornithidae' | ~ |
| Not 'Bohaiornithidae' | - |
| Not in study | × |

| | *Beiguornis* | *Bohaiornis* | *Longusunguis* | *Parabohaiornis* | *Shenqiornis* | *Sulcavis* | *Zhouornis* | *Gretcheniao* | *Musivavis* | *Eoenantiornis* | *Fortunguavis* | *Linyiornis* |
| --- | --- | --- | --- | --- | --- | --- | --- | --- | --- | --- | --- | --- |
| *Wang, 2014a* | × | + | + | + | + | + | + | × | × | ~ | × | × |
| *Wang et al., 2015c* | × | + | + | + | + | + | + | × | × | - | - | × |
| *Wang et al., 2015a* | × | + | + | + | + | + | + | × | × | ~ | - | × |
| *Wang and Liu, 2016a* | × | + | + | + | + | + | + | × | × | ~ | ~ | × |
| *Wang et al., 2016b* | × | + | + | + | + | + | + | × | × | ~ | + | + |
| *Hu and O'Connor, 2017* | × | + | + | + | + | + | + | × | × | - | + | × |
| *Wang and Zhou, 2017b* | × | + | + | + | + | + | + | × | × | - | ~ | ~ |
| *Cau, 2018* | × | × | × | × | × | + | × | × | × | × | × | × |
| *Chiappe et al., 2019* | × | + | + | + | - | - | - | - | × | - | - | × |
| *Zhang and Wang, 2019* | × | + | + | + | + | + | + | × | × | × | ~ | ~ |
| *Hu et al., 2020b* | × | + | + | - | + | + | - | × | × | - | - | - |
| *O'Connor et al., 2020b* | × | + | ~ | + | ~ | ~ | ~ | × | ~ | ~ | ~ | ~ |
| *Pittman, 2020b,* new technology | × | + | + | + | + | + | + | × | × | ~ | - | - |
| *Pittman, 2020b,* traditional | × | + | + | + | + | + | + | × | × | ~ | ~ | ~ |

*Table 1 continued*

| | Beiguornis | Bohaiornis | Longusunguis | Parabohaiornis | Shenqiornis | Sulcavis | Zhouornis | Gretcheniao | Musivavis | Eoenantiornis | Fortunguavis | Linyiornis |
|---|---|---|---|---|---|---|---|---|---|---|---|---|
| **Wang and Zhou, 2020b** | × | + | + | + | + | + | + | × | × | - | - | - |
| **Li et al., 2022,** strict consensus | × | + | ~ | + | ~ | ~ | ~ | × | × | ~ | × | × |
| **Li et al., 2022,** reduced consensus | × | + | + | + | + | + | + | × | × | ~ | × | × |
| **Liu et al., 2022** | × | + | + | - | + | + | + | ~ | × | - | - | × |
| **Wang et al., 2021b** | × | + | + | + | + | + | + | - | × | - | - | - |
| **Wang, 2022a** | + | + | - | + | + | + | + | - | × | + | + | × |
| **Wang et al., 2022b,** unweighted | × | + | - | × | + | + | + | - | - | + | + | × |
| **Wang et al., 2022b,** K = 20 | × | + | - | - | + | - | - | - | - | - | - | × |
| **Wang et al., 2022b,** K = 5 | × | + | - | - | + | - | + | + | - | + | - | × |

**Table 2.** Diet cut-offs used in this study.

Percentages refer to values given in EltonTraits 1.0 (*Wilman et al., 2014*), with Diet-Tetr being the sum of Diet-Ect and Diet-End (ectotherm and endotherm tetrapod food sources are combined). Granivores were separated into husking and swallowing subdivisions based on feeding descriptions in the literature.

| Diet | Cut-off |
| --- | --- |
| Folivore | 60+% Diet-PlantO |
| Frugivore | 60+% Diet-Fruit |
| Generalist | 40% or less in any category |
| Granivore | 70+% Diet-Seed |
| Invertivore | 60+% Diet-Inv |
| Nectarivore | 60+% Diet-Nect |
| Piscivore | 50+% Diet-Fish |
| Scavenger | 50+% Diet-Scav |
| Tetrapod Hunter | 60+% Diet-Tetr |

estimate (*Zhouornis* CNUVB-903) is an outlier. With the larger *Zhouornis* excluded, masses range from 91 to 406 g, x̄ = 248 g.

## Mechanical advantage and functional indices

Graphs of MA results are available in *Figure 4—figure supplement 2* (univariate) and *Figure 4* (multivariate), with 3D graphs in the data repository (https://doi.org/10.17632/7xtpbv27zh.3). Posterior predictions of bohaiornithid diet from flexible discriminate analysis (FDA) are provided in *Table 4*. Extant MA and functional index results are unchanged from *Miller et al., 2023*.

Bohaiornithids generally have low anterior jaw-closing mechanical advantage (AMA), posterior jaw-closing mechanical advantage (PMA), relative articular offset (AO), and relative maximum mandibular height (MMH) relative to living birds (*Figure 4—figure supplement 2A–D, G, H and J*). Most bohaiornithids also have a low jaw-opening mechanical advantage (OMA), but *Bohaiornis* has a high OMA. Other functional indices are intermediate across bohaiornithids.

In principal component analysis (PCA), *Longusunguis*, *Shenqiornis*, and *Sulcavis*, all plot near one another in a region inhabited by invertivores and generalists. This is driven by their low jaw-closing MA (*Figure 4—figure supplement 1A*). *Parabohaiornis*, with a slightly higher jaw-closing MA, plots nearby but closer to frugivores. *Bohaiornis* plots in an unoccupied region nearest frugivores and tetrapod hunters, separating from other bohaiornithids by its high upper jaw OMA (*Figure 4—figure supplement 1A*). *Zhouornis* plots between *Bohaiornis* and other bohaiornithids near invertivores and granivores due to its high relative average cranial height (ACH).

In flexible discriminant analysis (FDA), bohaiornithids other than *Bohaiornis* plot together in an indeterminate region inhabited by all diets but folivores and husking granivores. This mirrors their high affinity with generalist feeders (*Table 4*). *Bohaiornis* plots far from other bohaiornithids and the extant-inhabited area of the functional phylomorphospace along discriminant axis 2 (*Figure 4—figure supplement 1B*). The reason for this is unclear. The only functional index in which *Bohaiornis* differs from other bohaiornithids is OMA (*Figure 4—figure supplement 2*), which is loaded primarily on discriminant axis 1 (*Figure 4—figure supplement 1B*). Phylogenetic flexible discriminate analysis (pFDA) does not provide meaningful results when applied to functional index data, likely due to its poor explanatory power of the extant data (*Miller et al., 2023*).

Significant phylogenetic signal is present in the extant MA dataset overall (*Supplementary file 1*) and in each individual functional index (*Supplementary file 2*). Indices are usually less similar than expected under a BM model (K = 0.35–0.78), with the exception of relative ACH (K = 1.15). Phylogenetic HSD (i.e. comparison of means using the pairwise function in R package RRPP; *Collyer and Adams, 2018*) recovered no change in significant differences from HSD with the older phylogeny used

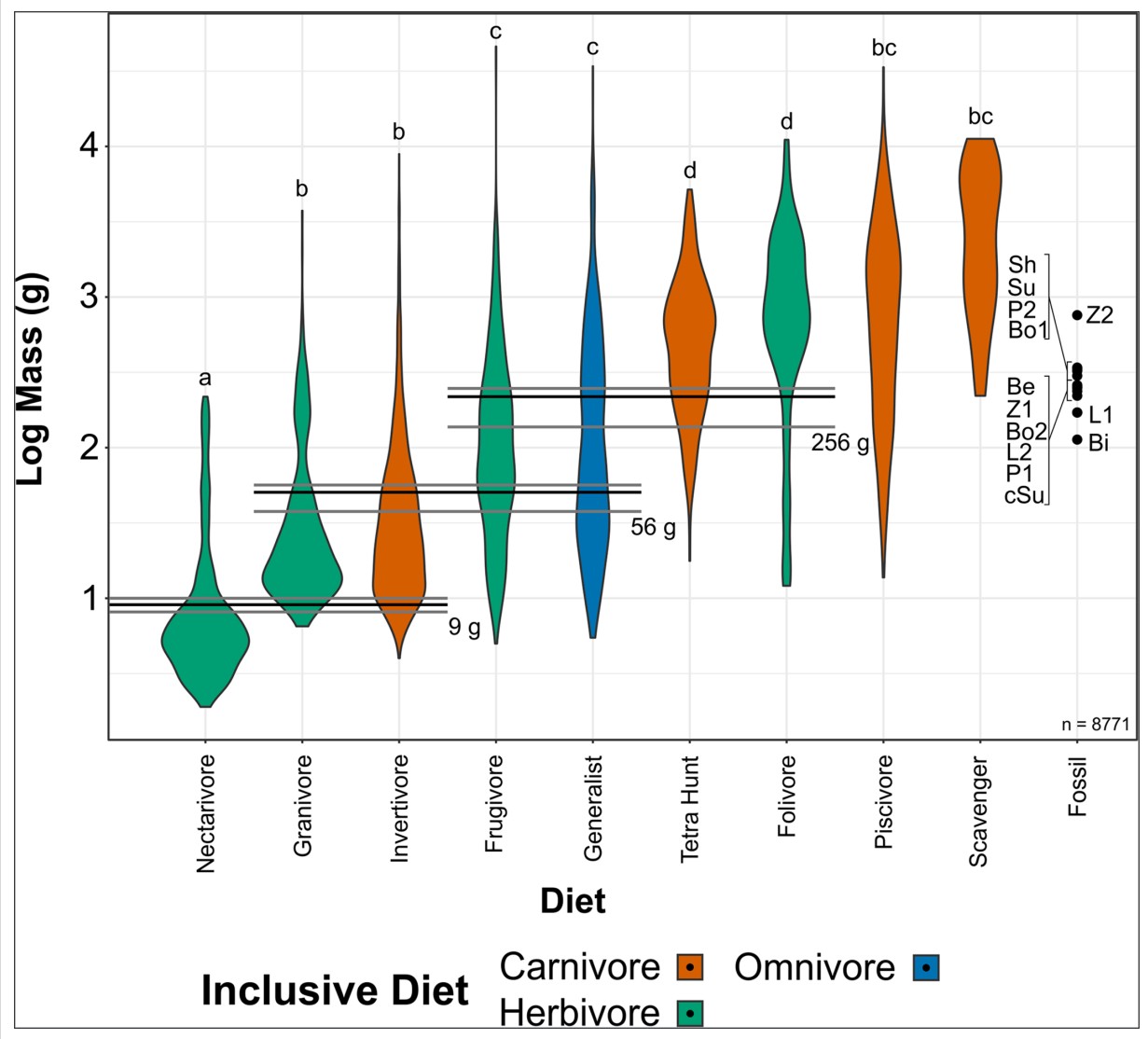

**Figure 3.** Violin plots of bird mass by diet, arranged in order of ascending mean mass. Masses were tested for significant differences via phylogenetic honest significant differences (HSD). Diets marked with the same letter are not significantly different from one another. Cut-off points between significantly different mass groups (black lines, with 95% CIs as grey lines) were calculated by optimising the Youden index and plotted. Note that, unlike in other diet treatments herein, granivores are not separated into husking and swallowing granivores. Mean bohaiornithid mass estimates are plotted for comparison, see *Table 3*. Diet abbreviations: Tetra Hunt, Tetrapod Hunter. Fossil taxon abbreviations: Be, *Beiguornis khinganensis* MHGU-F307/8; Bi, Bohaiornithidae indet. CUGB P1202; Bo1, *Bohaiornis* LPM B00167; Bo2, *Bohaiornis* IVPP V17963; L1, *Longusunguis* IVPP V17964; L2, *Longusunguis* IVPP V18693; P1, *Parabohaiornis* IVPP V18691; P2, *Parabohaiornis* IVPP V28398; Sh, *Shenqiornis* DNHM D2950/1; Su, *Sulcavis* BMNH Ph-805; cSu, cf. *Sulcavis* BMNHC-Ph1204; Z1, *Zhouornis* CNUVB-0903, Z2, *Zhouornis* BMNHC Ph 756.

in *Miller et al., 2023* (*Supplementary file 3*). Skull reconstructions used in MA and FEA calculations are provided in *Figure 2*.

Sensitivity analysis of the quadrate placement in fossil taxa (*Figure 2—figure supplement 1*) agrees with *Miller et al., 2022* and *Miller et al., 2023* that anterior shifts make folivory more likely and posterior shifts make piscivory more likely (*Supplementary file 4*). Scavenging was recovered as likely for *Parabohaiornis* and folivory was recovered as likely for *Bohaiornis* regardless of the quadrate's position.

### Finite element analysis

Graphs of FEA results are available in *Figures 5* (mesh-weighted arithmetic mean [MWAM] strain *Marcé-Nogué, 2016*) and *6* (multivariate strain intervals; *Marcé-Nogué et al., 2017*), with 3D graphs

**Table 3.** Masses for bohaiornithid taxa based on the regression equations of *Serrano et al., 2015*.
Most masses were previously reported in *Miller and Pittman, 2021*, though masses for the juvenile bohaiornithid CUGB P1202 (*Peteya et al., 2017*), *Beiguornis khinganensis* MHGU-F307/8 (*Wang, 2022a*), and cf. *Sulcavis* BMNHC-Ph1204 (*Liu et al., 2022*) are newly calculated in this study from literature images.

| Taxon | Specimen | Ontogenetic stage | Mean mass estimate (g) | Min mass estimate (g) | Max Mass Estimate (g) |
|---|---|---|---|---|---|
| Bohaiornithidae indet. | CUGB P1202 | 1 | 113 | 91 | 135 |
| *Beiguornis khinganensis* | MHGU-F307/8 | 2a | 260 | 210 | 310 |
| *Bohaiornis guoi* | IVPP V17963 | 3c | 300 | 242 | 358 |
| *B. guoi* | LPM B00167 | 2b | 249 | 201 | 298 |
| *Longusunguis kurochkini* | IVPP V17964 | 2b | 171 | 137 | 204 |
| *L. kurochkini* | IVPP V18693 | 3a | 237 | 191 | 283 |
| *Parabohaiornis martini* | IVPP V18691 | 2b | 221 | 178 | 263 |
| *Parabohaiornis martini* | IVPP V28398 | 3b | 323 | 260 | 386 |
| *Shenqiornis mengi* | DNHM D2950/1 | 2a | 340 | 274 | 406 |
| *Sulcavis geeorum* | BMNHC-Ph805 | 3b | 333 | 268 | 397 |
| cf. *Sulcavis* | BMNHC-Ph1204 | 2a | 171 | 137 | 204 |
| *Zhouornis hani* | BMNHC-Ph756 | 2a | 253 | 204 | 303 |
| *Z. hani* | CNUVB-903 | 3c | 758 | 611 | 905 |

in the data repository (https://doi.org/10.17632/7xtpbv27zh.3). Posterior predictions of bohaiornithid diet from FDA are provided in *Table 5*. Extant FEA results are unchanged from *Miller et al., 2023*.

MWAM strain of bohaiornithids is low (89–156 µε, x̄ = 128 µε), with *Bohaiornis* and *Zhouornis* experiencing less strain under loading that any extant carnivorous bird in this study (min = 105 µε).

In PCA, *Bohaiornis* inhabits a region of the strain-space that is only inhabited by extant herbivores and *Zhouornis* inhabits a space that is exclusively non-carnivorous. *Longusunguis*, *Parabohaiornis*, and *Sulcavis* inhabit a region where all diets overlap. *Shenqiornis* inhabits a nearby region with slightly less occupation by invertivores, piscivores, and frugivores. Bohaiornithids tend to display heterogeneous strain, that is, more model area under high or low strain rather than under intermediate strain.

In FDA, bohaiornithids tend to plot in uninhabited areas of the strain-space (*Figure 6B*). Thus, the high confidence of posterior predictions from FEA (*Table 5*) reflects the nearest extant diet group. As previously noted for this dataset (*Miller et al., 2022*), groups with similar distance from the origin are not meaningfully different in jaw strength, so all bohaiornithids, not just specific taxa, should be interpreted as plotting in the same functional space as folivores, frugivores, nectarivores, piscivores, and scavengers. The exception is *Longusunguis*, the bohaiornithid with the second weakest jaw whose affinities are strongest with generalist feeders. pFDA does not provide meaningful results when applied to FEA intervals data, likely due to the low phylogenetic signal in these data (*Miller et al., 2023*).

No significant phylogenetic signal was recovered in the extant FEA data (*Supplementary file 1*). Phylogenetic HSD does recover less significant differences in diet than *Miller et al., 2023* when using the updated extant phylogeny. We no longer find significant differences between folivores and husking granivores or scavengers, between frugivores and scavengers, or between husking granivores and invertivores (*Supplementary file 5*).

## Traditional morphometrics

### Pes

Graphs of pedal TM results are available in *Figure 7*, with character weights in *Figure 7—figure supplement 1* and 3D graphs in the data repository (https://doi.org/10.17632/7xtpbv27zh.3). Posterior predictions of bohaiornithid pedal ecology from FDA and pFDA are provided in *Table 6*. Extant pedal TM results are almost unchanged from *Miller et al., 2023*, with minor differences from added data noted below. Bohaiornithids generally spread across the phylomorphospace. In PCA, *Sulcavis*

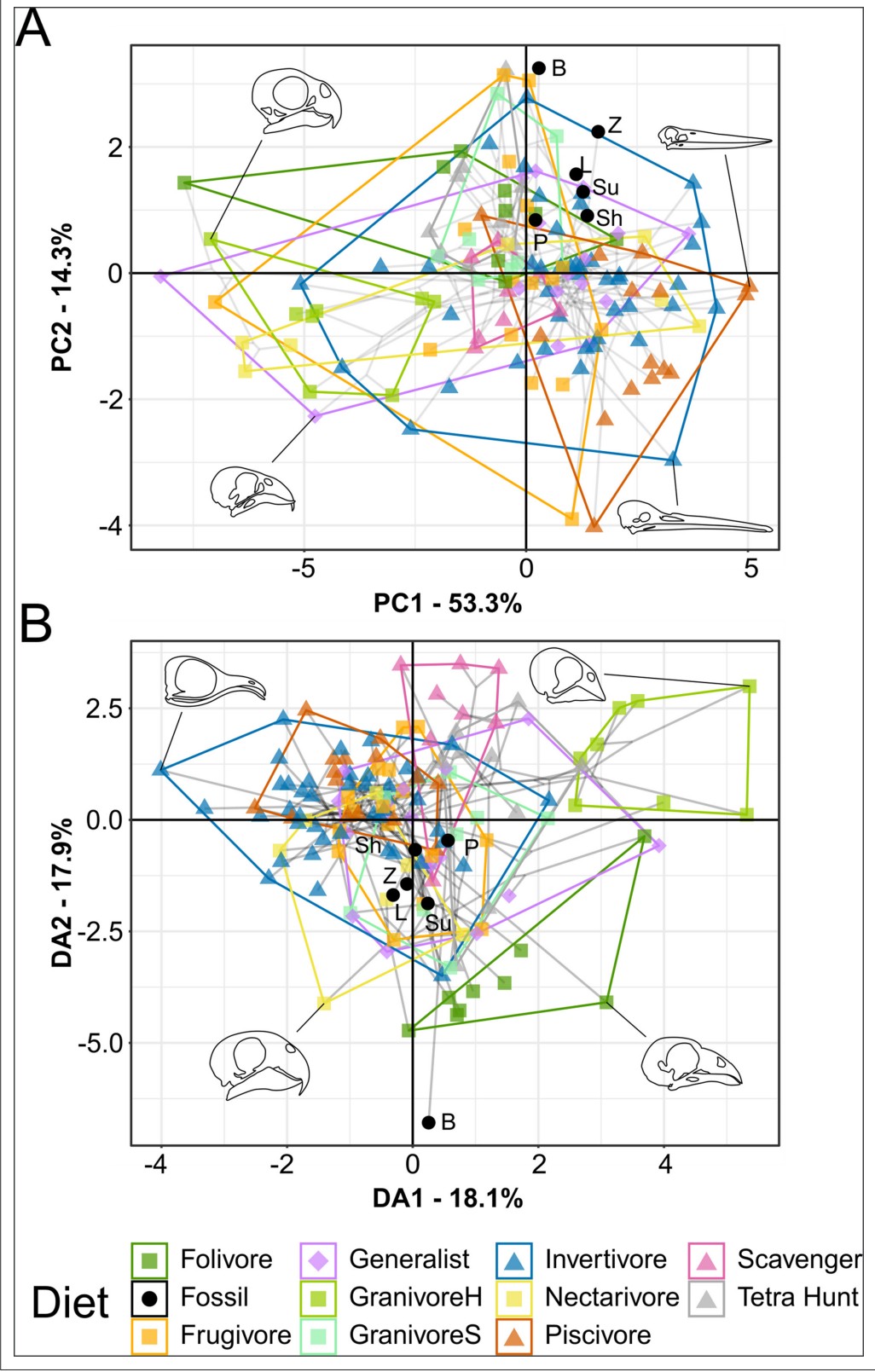

**Figure 4.** Functional phylomorphospace of mechanical advantage (MA) and functional index data, grouped by diet. Grey lines indicate phylogenetic relationships. Line drawings of skulls for selected taxa are provided for reference. Data are presented with principal component analysis (PCA) (**A**) and flexible discriminant analysis (FDA) (**B**). See *Figure 4—figure supplement 1* for character weights and *Table 4* for FDA posterior predictions. Diet

*Figure 4 continued on next page*

*Figure 4 continued*
abbreviations: GranivoreH, Husking Granivore; GranivoreS, Swallowing Granivore; Tetra Hunt, Tetrapod Hunter. Fossil taxon abbreviations: B, *Bohaiornis*; L, *Longusunguis*; P, *Parabohaiornis*; Sh, *Shenqiornis*; Su, *Sulcavis*; Z, *Zhouornis*.

The online version of this article includes the following figure supplement(s) for figure 4:

**Figure supplement 1.** Plot of character weightings for the graphs in *Figure 4*.

**Figure supplement 2.** Violin plots of individual functional indices, organised by diet.

plots among large raptors and both *Parabohaiornis* individuals plot among non-raptorial perching birds (though the latter separate into an adjacent uninhabited space with PC3 included, see data repository). *Longusunguis* IVPP V18693, both specimens of *Bohaiornis*, and both specimens of *Zhouornis* plot in an indeterminate region of primarily small raptors and non-raptorial perching birds. *Longusunguis* in particular also plots near tinamous, which are ground birds.

In FDA, *Parabohaiornis* shows a consistent affinity with non-raptorial perching birds (*Table 6*). Both specimens of *Bohaiornis* and the younger specimen of *Zhouornis* show affinity with small raptors as well as non-raptorial perching birds, with the older *Zhouornis* showing more affinity with ground birds than perching birds (*Table 6*). *Longusunguis* and *Sulcavis* both show affinity with large raptors, with *Longusunguis* also similar to non-raptorial perchers and *Sulcavis* to small raptors (*Table 6*).

While pFDA results differ from *Miller et al., 2023* due to the use of a new phylogeny and additional extant data (see 'Methods'), the differences are minimal. The new optimal $\lambda$ of the extant data is 0.33 (optimal $\lambda$ was 0.31 in *Miller et al., 2023*), and axis loadings are nearly identical (*Figure 7—figure supplement 1C*). In pFDA, *Bohaiornis* LPM B00167 shows affinity with small raptors, *Longusunguis* with large raptors, and *Zhouornis* CNUVB-903 with non-raptorial perching birds (*Table 6*). The remaining bohaiornithid fossils plot outside the phylomorphospace occupied by extant taxa (*Figure 7C*).

Significant phylogenetic signal is present in the extant pedal TM dataset overall (*Supplementary file 1*) and in each individual measurement (*Supplementary file 6*). Measurements are all less similar than expected under a BM model (K = 0.31–0.78), with measurements involving digits I and II having higher K values. Phylogenetic HSD recovered no change in significant differences from HSD with the older phylogeny used in *Miller et al., 2023* (*Supplementary file 7*).

The additional extant data has produced little change from *Miller et al., 2023*, primarily extending the morphospace of non-raptorial perching birds into more negative PC1 space (*Figure 7A*) driven by the inclusion of the tawny frogmouth *Podargus strigoides*.

## Skull

A graph of skull TM as described in its original application to enantiornithines (*Clark et al., 2023*) is available in *Figure 8—figure supplement 1A*. However, as seen in *Figure 8—figure supplement 1B* and discussed at length in 'Methods', skull length appears to be a size proxy that is redundant given the mass data examined in this work. Instead, we compare relative rostrum length as used in *Clark et al., 2023* to relative skull length (i.e. skull length normalised to body size; *Figure 8*).

Granivores and tetrapod hunters tend to have relatively short rostra (rostrum length/skull length <0.8) while folivores, nectarivores, and piscivores tend to have more elongate rostra (rostrum length/skull length >0.8). Frugivores, generalists, and invertivores display a variety of rostral proportions.

Nectarivores have distinctly elongate skulls (relative skull length >4.5), though this is exclusively true for hummingbirds (Trochilidae). Somewhat elongate skulls (relative skull length >3.5) are primarily found in invertivores and generalists, though these diet groups overall display a range of relative skull lengths.

Bohaiornithids display distinctly short skulls (relative skull length <1.6), with *Zhouornis* displaying the shortest skull relative to body mass of any taxon sampled. Their relative rostrum lengths fall in the middle of the extant range.

Significant phylogenetic signal is present in the skull TM data. When using the data as is, a K value near zero is returned, driven by the variation observed within the taxa sampled multiple times. While this has interesting implications for the importance of individual variation in morphometric studies, for simplicity in this work we averaged the traits of all species with multiple samples when calculating phylogenetic signal. Skull and rostrum length distribution both resemble a BM model (K = 0.89 and

**Table 4.** Posterior probabilities predicting bohaiornithid diet by flexible discriminate analysis (FDA) from mechanical advantage (MA) and functional indices of extant bird jaws.

Values in blue are most likely, values in red are least likely. All bohaiornithids have high affinity with generalists and low affinity with husking granivores, with other affinities varying by taxon. Diet abbreviations: GranivoreH, Husking Granivore; GranivoreS, Swallowing Granivore; Tetra Hunt, Tetrapod Hunter.

| Taxon | Folivore | Frugivore | Generalist | GranivoreH | GranivoreS | Invertivore | Nectarivore | Piscivore | Scavenger | Tetra Hunt |
|---|---|---|---|---|---|---|---|---|---|---|
| Bohaiornis | 9.99E-01 | 6.43E-06 | 2.92E-04 | 5.97E-13 | 2.15E-04 | 5.43E-06 | 1.62E-05 | 2.00E-07 | 1.74E-10 | 7.78E-08 |
| Longusunguis | 1.53E-02 | 1.81E-01 | 5.39E-01 | 5.48E-07 | 4.38E-03 | 1.57E-01 | 3.05E-02 | 6.91E-02 | 9.85E-05 | 3.95E-03 |
| Parabohaiornis | 2.44E-01 | 6.72E-04 | 2.34E-01 | 2.49E-07 | 3.64E-01 | 1.05E-02 | 2.42E-04 | 1.49E-02 | 1.32E-01 | 7.69E-05 |
| Shenqiornis | 7.32E-03 | 1.12E-02 | 6.45E-01 | 1.81E-06 | 2.00E-03 | 5.15E-02 | 8.12E-03 | 2.65E-01 | 9.50E-03 | 5.24E-04 |
| Sulcavis | 7.28E-02 | 1.58E-02 | 8.22E-01 | 5.46E-07 | 4.12E-03 | 2.43E-02 | 3.39E-03 | 5.68E-02 | 5.74E-04 | 6.25E-04 |
| Zhouornis | 3.20E-02 | 1.61E-01 | 3.11E-01 | 9.82E-07 | 2.30E-02 | 2.69E-01 | 5.42E-02 | 7.86E-02 | 9.64E-04 | 6.99E-02 |

**Table 5.** Posterior probabilities predicting bohaiornithid diet by flexible discriminate analysis (FDA) from finite element analysis (FEA) following the intervals method (*Marcé-Nogué et al., 2017*).

Values in blue are most likely, values in red are least likely. Bohaiornithid affinities varying considerably between taxa, only universally not resembling tetrapod hunters. Diet abbreviations: GranivoreH, Husking Granivore; GranivoreS, Swallowing Granivore; Tetra Hunt, Tetrapod Hunter.

| Taxon | Folivore | Frugivore | Generalist | GranivoreH | GranivoreS | Invertivore | Nectarivore | Piscivore | Scavenger | Tetra Hunt |
|---|---|---|---|---|---|---|---|---|---|---|
| Bohaiornis | 3.44E-10 | 9.33E-12 | 1.73E-07 | 2.25E-23 | 1.00E+00 | 7.09E-07 | 4.67E-19 | 1.43E-10 | 8.82E-06 | 3.37E-19 |
| Longusunguis | 3.05E-01 | 1.15E-13 | 6.95E-01 | 1.56E-26 | 2.26E-15 | 3.58E-09 | 4.35E-26 | 1.51E-11 | 3.47E-29 | 1.07E-20 |
| Parabohaiornis | 8.49E-02 | 3.30E-15 | 2.08E-13 | 3.50E-24 | 5.52E-01 | 1.18E-04 | 4.67E-04 | 3.63E-01 | 3.12E-13 | 1.13E-06 |
| Shenqiornis | 2.28E-08 | 1.45E-23 | 9.39E-09 | 4.31E-11 | 2.28E-13 | 2.47E-05 | 3.61E-02 | 9.58E-01 | 6.11E-03 | 2.81E-20 |
| Sulcavis | 6.82E-06 | 1.91E-16 | 1.19E-03 | 9.94E-01 | 5.99E-18 | 5.65E-09 | 4.71E-03 | 1.22E-07 | 4.68E-35 | 1.55E-22 |
| Zhouornis | 5.69E-50 | 1.00E+00 | 5.55E-20 | 4.89E-13 | 4.06E-12 | 2.08E-18 | 8.99E-13 | 4.74E-20 | 1.13E-53 | 2.52E-17 |

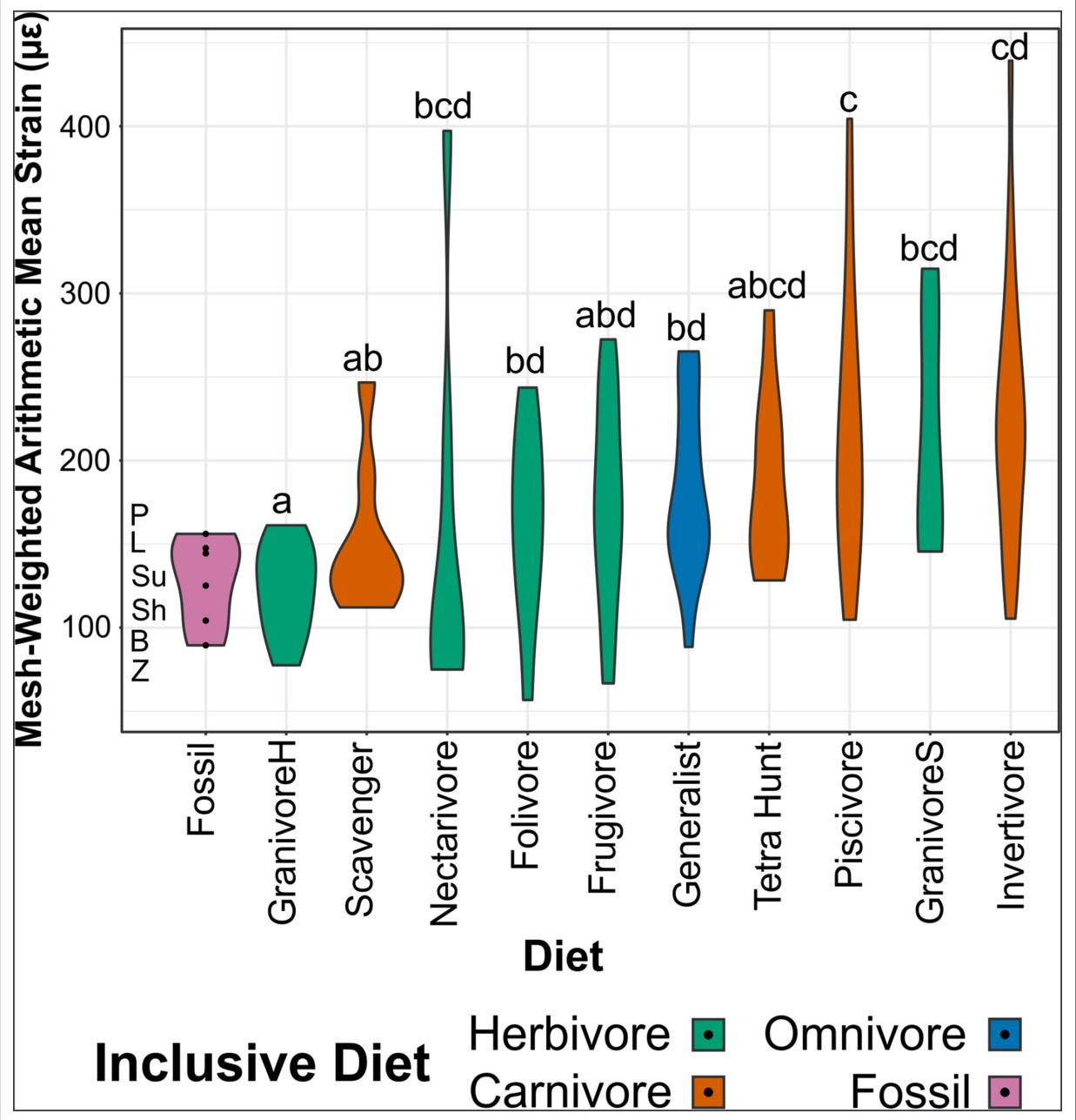

**Figure 5.** Violin plots of mesh-weighted arithmetic mean (MWAM) strain of finite element analysis (FEA) models, organised by diet. Extant diets ascend in average MWAM strain from left to right. MWAM strains were tested for significant differences via phylogenetic honest significant differences (HSD). Diets marked with the same letter are not significantly different from one another. Diet abbreviations: GranivoreH, Husking Granivore; GranivoreS, Swallowing Granivore; Tetra Hunt, Tetrapod Hunter. Fossil taxon abbreviations: B, *Bohaiornis*; L, *Longusunguis*; P, *Parabohaiornis*; Sh, *Shenqiornis*; Su, *Sulcavis*; Z, *Zhouornis*.

0.91), while the ratio between them is less similar than expected under BM (K = 0.64) and relative skull length is more similar (K = 1.31; ***Supplementary file 8***).

## Ancestral state reconstruction

Qualitative ancestral state reconstruction of enantiornithine diet is presented in ***Figure 9***. Precise values for nodes of interest in ***Figure 9*** are provided here. The common ancestor of Avisauridae is recovered as 100% likely to be a vertivore. The common ancestor of Bohaiornithidae is recovered as 34% likely to be an herbivore, 33% likely to be a vertivore, and 33% likely to be an omnivore. The

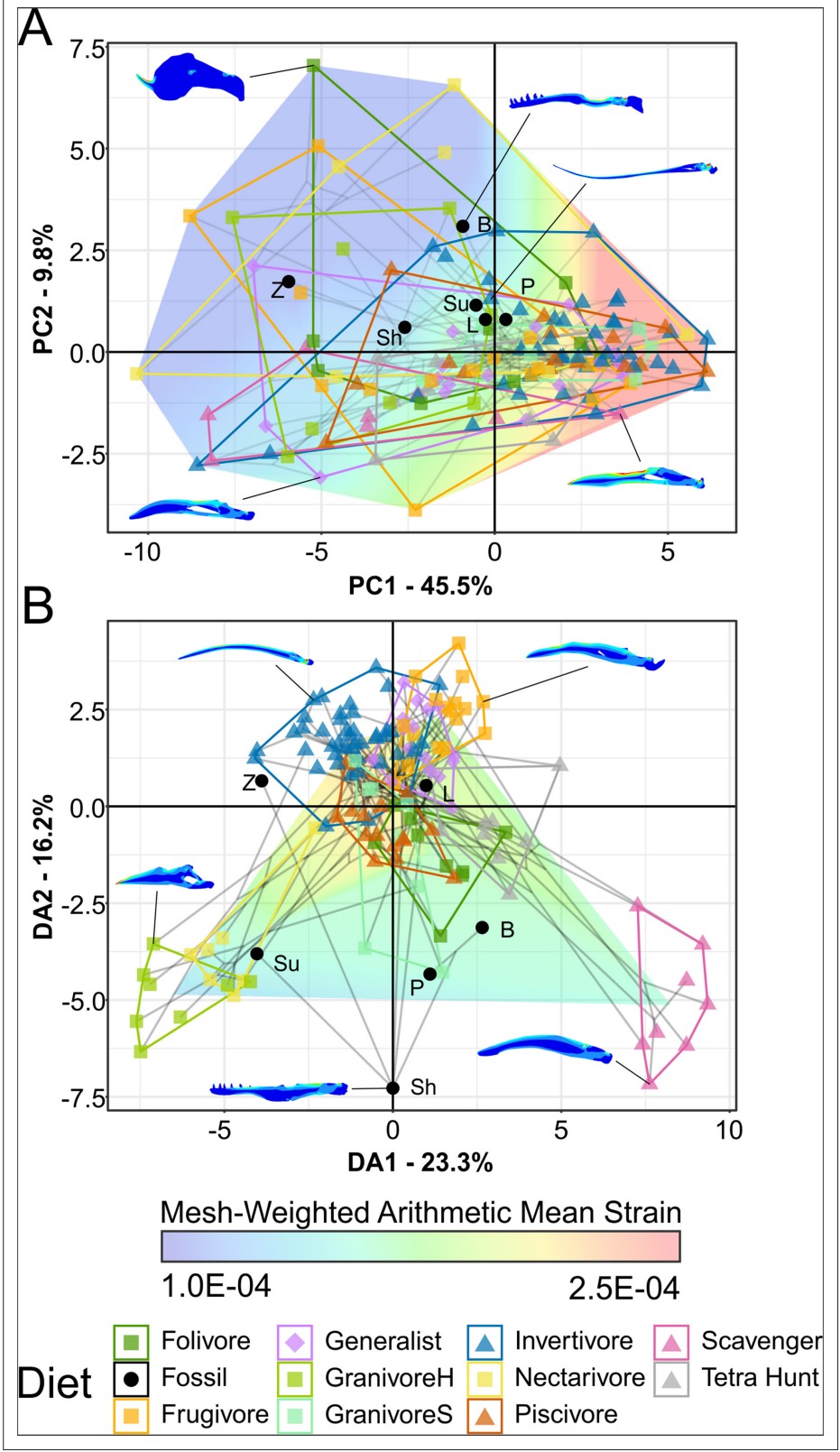

**Figure 6.** Phylogenetic strain-space of total maximum in-plane principal strain intervals for extant and fossil bird lower jaw finite element models, grouped by diet. Mesh-weighted arithmetic mean (MWAM) strain is mapped overtop the data. Grey lines indicate phylogenetic relationships. Contour plots for selected taxa are provided for reference. Data are presented with principal component analysis (PCA) (**A**) and flexible discriminate analysis (FDA)

*Figure 6 continued on next page*

*Figure 6 continued*
(**B**). In PCA (**A**), overall strain increases along PC1, and strain heterogeneity (i.e. lower areas of intermediate strain) increases along PC2. In FDA (**B**), DA1 and DA2 have loadings of various similar low-strain intervals, with high-strain intervals clustering near the origin. See *Table 5* for FDA posterior predictions. Diet abbreviations: GranivoreH, Husking Granivore; GranivoreS, Swallowing Granivore; Tetra Hunt, Tetrapod Hunter. Fossil taxon abbreviations: B, *Bohaiornis*; L, *Longusunguis*; P, *Parabohaiornis*; Sh, *Shenqiornis*; Su, *Sulcavis*; Z, *Zhouornis*.

---

common ancestor of Longipterygidae is recovered as 100% likely to be an invertivore. The diet of the common ancestor of Pengornithidae is recovered as 99% unknown, with the next most likely diet invertivory at 0.02%. The same is true for the common ancestor of all Enantiornithes. The common ancestor of *Linyiornis* and a large group of late-diverging enantiornithines is recovered as 99% likely to be an invertivore (1% unknown), and this is true for the rest of the backbone of the tree crownward from this point. Numerical likelihoods of each node are available in the data repository (https://doi.org/10.17632/7xtpbv27zh.3).

Quantitative ancestral states are provided for body mass (*Figure 9—figure supplement 1*), MA and functional indices (*Figure 9—figure supplement 2*), FEA MWAM strain (*Figure 9—figure supplement 3*), and pedal TM (*Figure 9—figure supplement 4*). Due to software limitations, these presentations have polytomies non-randomly resolved and thus are only one possible reconstruction. *Table 7* provides more valid values of each quantitative trait for the common ancestor of Enantiornithes by averaging results from 10,000 random polytomy resolutions. To facilitate interpreting multivariate traits of the common ancestor of Enantiornithes, FDA posterior probabilities are provided in *Table 8* based on the common ancestor MA data in *Table 7*, and a PCA graph of the common ancestor pedal TM data is provided in *Figure 7—figure supplement 2*.

## Discussion

### Body mass

An increased sample size has improved resolution of the relationship between mass and diet over our previous work (*Miller et al., 2022*; *Miller et al., 2023*). Notably, the separation between vertivores and invertivores is at a lower mass (324–429 g in *Miller et al., 2022*; *Miller et al., 2023* vs 80–148 g here). This is of particular note for interpreting longipterygid diet. In a previous study (*Miller et al., 2022*), we found *Longipteryx* unlikely to be a piscivore due to its low body mass, but three specimens of *Longipteryx* (DNHM D2889, est. 154 g; IVPP V12325, est. 193 g; and STM 8–117, est. 206 g) are above the mass cut-off in this study. While MA results do still indicate piscivory not being particularly likely in *Longipteryx* (*Miller et al., 2022*), we believe that with this refined mass data the hypothesis of *Longipteryx* as a specialist piscivore (*O'Connor and Chiappe, 2011a*; *Chiappe and Meng, 2016*; *Zhang et al., 2001*; *Wang et al., 2015b*; *O'Connor, 2009*; *Martyniuk, 2012*; *Chatterjee, 2015*; *Benito and Olivé, 2022*) can no longer be rejected. Specific analogy to kingfishers, whose jaw strain is lower than *Longipteryx* under loading, is still rejected.

Bohaiornithid masses are most consistent with folivory, frugivory, generalist feeding, piscivory, scavenging, or tetrapod hunting. Mean estimates of mass for most bohaiornithid specimens (*Table 3*) are near the 265 g split between frugivores + generalists + piscivores + scavengers (group c) and folivores + tetrapod hunters (group d) (*Figure 3*). The uncertainties of both the split and the masses cause all but two bohaiornithid specimens to fall within both groups c and d. The exceptions are the indefinite bohaiornithid CUGB P1202, firmly within group c but also highly immature, and *Zhouornis* CNUVB-903 whose estimated mass is firmly within group d. The largest specimens of *Bohaiornis*, *Shenqiornis*, *Sulcavis*, and *Zhouornis* have mean mass estimates which are greater than 265 g, which we interpret as stronger affinity with folivory and tetrapod hunting than other bohaiornithids.

We note that every bohaiornithid is less massive than any avian obligate scavenger, and facultative scavenging in birds as small as even the largest bohaiornithid, *Zhouornis* CNUVB-903, is facilitated by anthropogenic waste (*Thomson, 2016*; *Mazumdar et al., 2018*) and thus may not represent a natural state. Coupled with past work proposing very large body size is a prerequisite for obligate scavenging (*Ruxton and Houston, 2004*), we consider scavenging less likely in bohaiornithids than the other diets mentioned above.

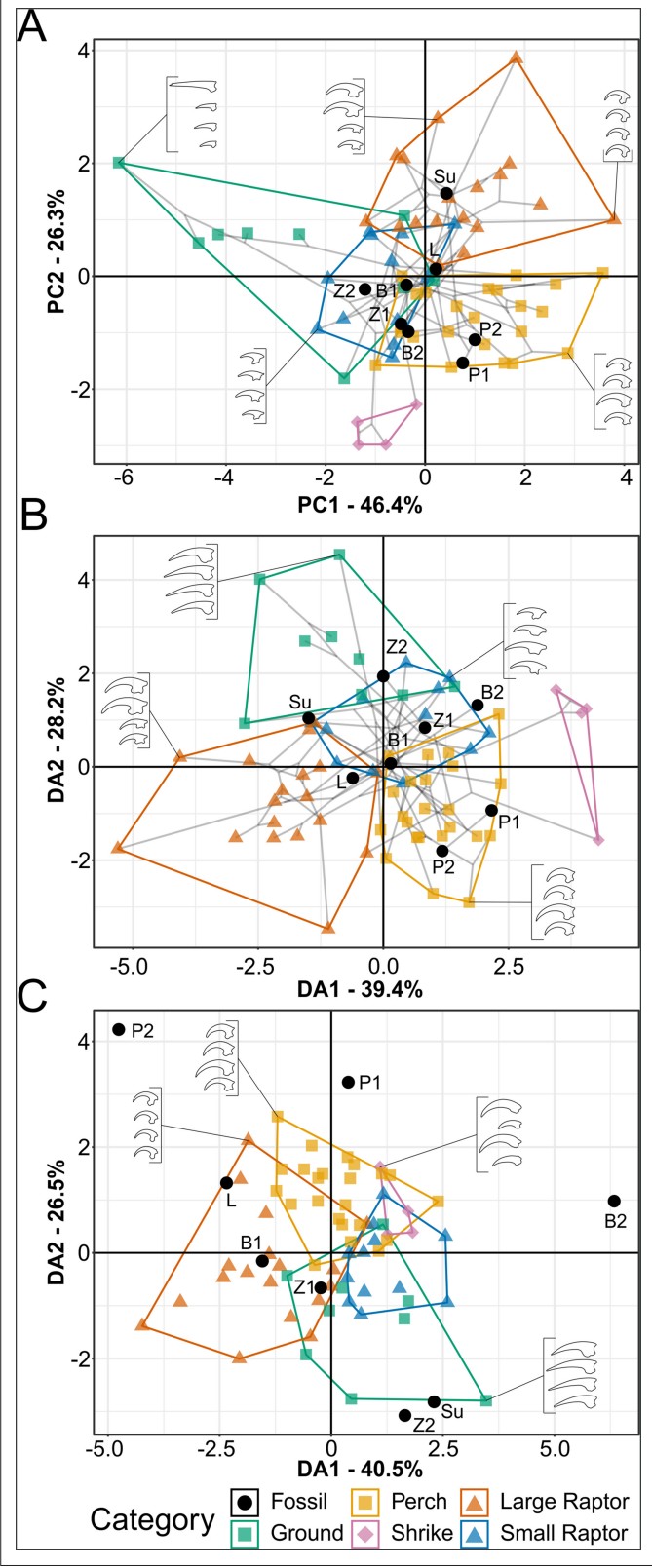

**Figure 7.** Phylomorphospace of extant and fossil bird claw shape from pedal traditional morphometric (TM), grouped by pedal ecological category. Grey lines indicate phylogenetic relationships. Line drawings of claws for selected taxa are provided for reference. Data are presented with principal component analysis (PCA) (**A**), flexible discriminate analysis (FDA) (**B**), and phylogenetic flexible discriminate analysis (pFDA) (**C**). See *Figure 7—figure*

*Figure 7 continued on next page*

*Figure 7 continued*

**supplement 1** for character weights and *Table 6* for FDA and pFDA posterior predictions. Category abbreviations: large raptor, raptor taking prey which does not fit in the foot; small raptor, raptor taking prey which can fit in the foot. Fossil taxon abbreviations: B1, *Bohaiornis* LPM B00167; B2, *Bohaiornis* IVPP V17963; L, *Longusunguis* IVPP V18693; P1, *Parabohaiornis* IVPP V18690; P2, *Parabohaiornis* IVPP V18691; Su, *Sulcavis* BMNH Ph-805; Z1, *Zhouornis* CNUVB-0903, Z2, *Zhouornis* BMNHC Ph 756.

The online version of this article includes the following figure supplement(s) for figure 7:

**Figure supplement 1.** Plot of character weightings for the graphs in *Figure 7*.

**Figure supplement 2.** Phylomorphospace of extant bird claw shape from traditional morphometric (TM), grouped by pedal ecological category, for comparison to the common ancestor of Enantiornithes.

## Mechanical advantage and functional indices

As previously observed (*Miller et al., 2022*; *Miller et al., 2023*), skull functional indices only effectively separate folivores, husking granivores, and scavengers in the functional morphospace (*Figure 4*). This means dietary resolution is poor from this line of evidence. In addition to previously proposed explanations of MA adaptations being only impactful in particular lineages (*Navalón et al., 2019*) or ecologies (*Miller et al., 2022*), we note here the need to investigate how cranial kinesis in extant birds affects measures of their skull function. Kinesis may modify both upper jaw anterior jaw-closing mechanical advantage (AMA) and jaw-opening mechanical advantage (OMA), which could also have

**Table 6.** Posterior probabilities predicting bohaiornithid pedal ecology by flexible discriminate analysis (FDA) and phylogenetic flexible discriminate analysis (pFDA) from traditional morphometric (TM) of extant bird claws.

Values in blue are most likely, values in red are least likely. Bohaiornithid affinities vary considerably by taxon and between FDA and pFDA. Diet abbreviations: GranivoreH, Husking Granivore; GranivoreS, Swallowing Granivore; Tetra Hunt, Tetrapod Hunter. Category abbreviations: large raptor, raptor taking prey which does not fit in the foot; small raptor, raptor taking prey which can fit in the foot.

| Test | Taxon | Ground | Perch | Large raptor | Small raptor | Shrike |
|---|---|---|---|---|---|---|
| | *Bohaiornis* LPM B00167 | 2.45E-02 | 4.03E-01 | 2.69E-01 | 3.03E-01 | 4.60E-04 |
| | *Bohaiornis* IVPP V17963 | 1.92E-02 | 2.52E-01 | 2.31E-04 | 7.23E-01 | 5.78E-03 |
| | *Longusunguis* | 3.65E-02 | 4.93E-01 | 3.76E-01 | 9.45E-02 | 4.42E-07 |
| | *Parabohaiornis* IVPP V18690 | 2.94E-05 | 8.57E-01 | 1.49E-03 | 5.57E-02 | 8.63E-02 |
| | *Parabohaiornis* IVPP V18691 | 2.41E-05 | 9.66E-01 | 2.66E-02 | 4.41E-03 | 2.85E-03 |
| | *Sulcavis* | 2.03E-01 | 1.80E-02 | 2.17E-01 | 5.62E-01 | 2.93E-09 |
| | *Zhouornis* CNUVB-0903 | 1.22E-01 | 3.62E-01 | 6.10E-02 | 4.07E-01 | 4.83E-02 |
| FDA | *Zhouornis* BMNHC Ph 756 | 5.46E-01 | 1.41E-02 | 1.72E-02 | 4.22E-01 | 1.33E-04 |
| | *Bohaiornis* LPM B00167 | 0.00E+00 | 6.09E-02 | 1.29E-01 | 8.10E-01 | 4.57E-05 |
| | *Bohaiornis* IVPP V17963 | 0.00E+00 | 6.21E-07 | 4.58E-06 | 4.87E-12 | 1.00E+00 |
| | *Longusunguis* | 0.00E+00 | 2.66E-03 | 9.94E-01 | 2.82E-03 | 6.46E-04 |
| | *Parabohaiornis* IVPP V18690 | 0.00E+00 | 1.98E-04 | 2.10E-01 | 7.90E-01 | 3.27E-12 |
| | *Parabohaiornis* IVPP V18691 | 0.00E+00 | 2.68E-02 | 3.05E-01 | 6.68E-01 | 3.28E-07 |
| | *Sulcavis* | 0.00E+00 | 2.87E-04 | 3.92E-05 | 3.27E-05 | 1.00E+00 |
| | *Zhouornis* CNUVB-0903 | 0.00E+00 | 4.94E-01 | 2.84E-01 | 2.19E-01 | 3.00E-03 |
| pFDA | *Zhouornis* BMNHC Ph 756 | 0.00E+00 | 3.10E-01 | 5.23E-03 | 3.96E-03 | 6.80E-01 |

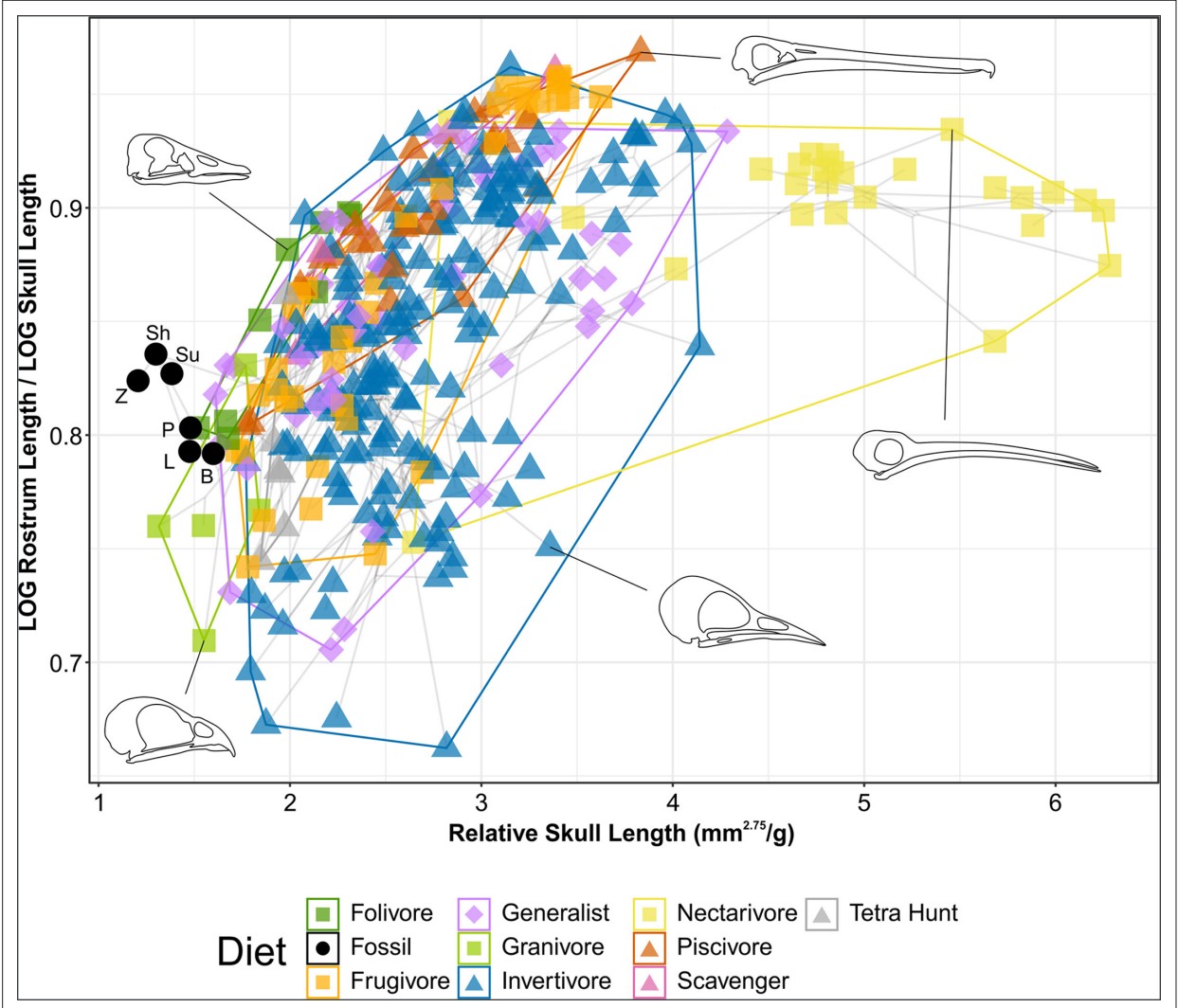

**Figure 8.** Phylomorphospace of extant and fossil bird skull proportions, grouped by diet. Grey lines indicate phylogenetic relationships. Line drawings of skulls for selected taxa are provided for reference. The data presented are modified from *Clark et al., 2023*, see *Figure 8—figure supplement 1* for data more directly comparable to that study. Diet abbreviation: Tetra Hunt, Tetrapod Hunter. Fossil taxon abbreviations: B, *Bohaiornis*; L, *Longusunguis*; P, *Parabohaiornis*; Sh, *Shenqiornis*; Su, *Sulcavis*; Z, *Zhouornis*.

The online version of this article includes the following figure supplement(s) for figure 8:

**Figure supplement 1.** Alternate phylomorphospaces to *Figure 8*.

consequences for the performance of the lower jaw. For example, the selective pressure for speed in the lower jaw may be reduced if rapid movement of the premaxilla also aids in prey acquisition. The need to increase rostral length to increase gape may also be reduced by movement of the upper beak. Unfortunately, the extent of excursion of the upper beak in, life and whether it occurs during biting, is unrecorded for the vast majority of taxa. Future investigation into this unknown and incorporation of the data into MA analyses may yield better explanatory power for ecological aspects.

Bohaiornithids separate into two distinct functional guilds: *Bohaiornis* and *Parabohaiornis* in one with distinctly high OMA; and *Longusunguis*, *Shenqiornis*, *Sulcavis*, and *Zhouornis* with OMA values more in line with the average extant bird. The posteroventral extreme of the cranium is a landmark location for OMA, and this region is reconstructed in every other taxon with material from *Bohaiornis* specimen IVPP V17963 (*Figure 2*), so it is possible that the OMA is overestimated in other bohaiornithid taxa. But given the similarity of the posterior deflection of the posterodorsal cranium in *Shenqiornis*, *Sulcavis*, and especially *Parabohaiornis* to the deflection seen in *Bohaiornis,* we consider OMA values for non-*Bohaiornis* bohaiornithids to be reasonable estimates.

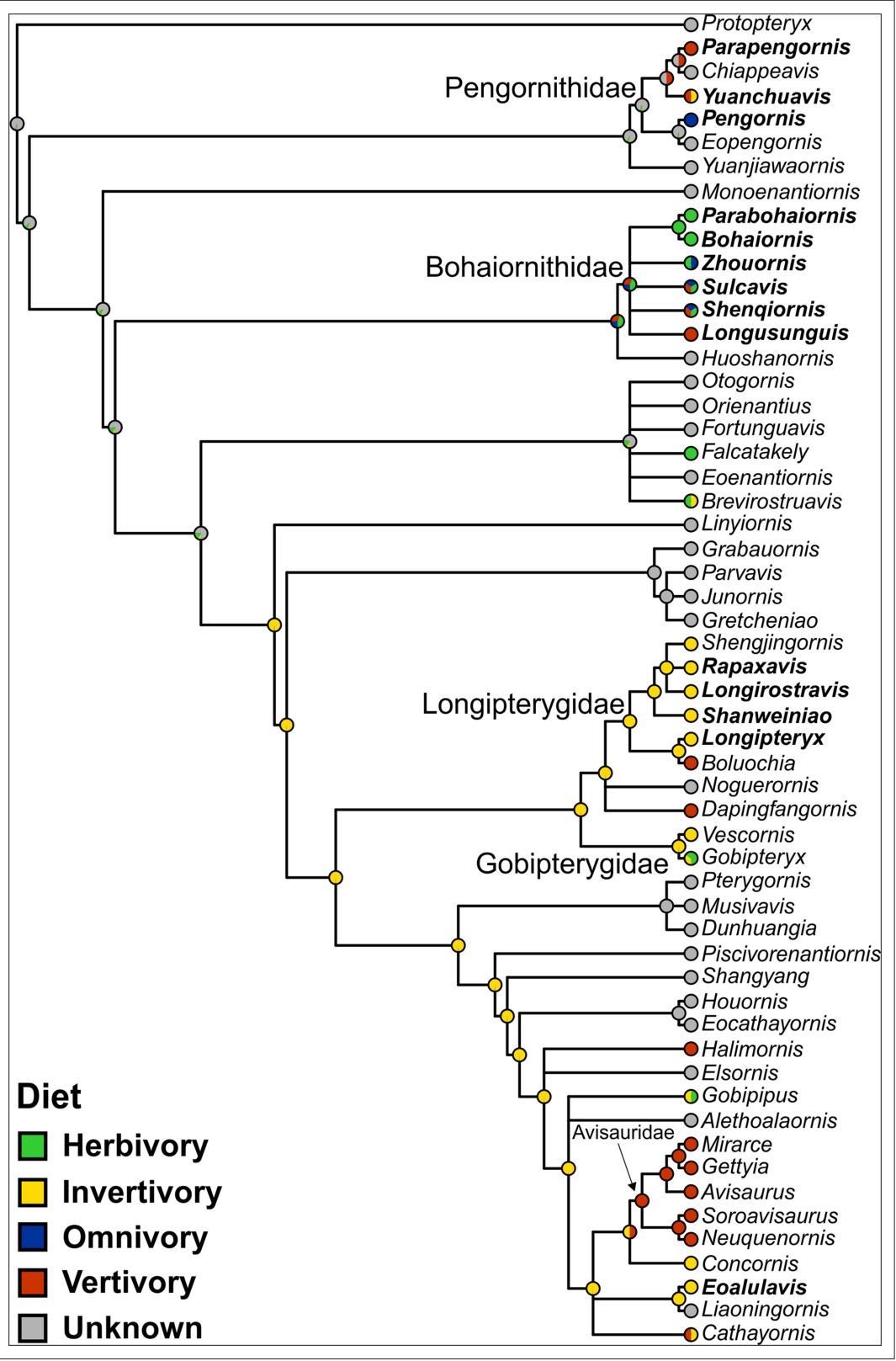

**Figure 9.** Ancestral state reconstruction of enantiornithine diet. Phylogeny is presented as not time-scaled for node visibility. All enantiornithine taxa with dietary hypotheses are included, as well as enantiornithines complete enough to create robust mass estimates (*Miller and Pittman, 2021*; *Miller et al., 2022*; *Miller et al., 2023*; *Serrano et al., 2015*) and the non-ornithothoracine pygostylian *Sapeornis* as an outgroup. Taxa with bold names

*Figure 9 continued on next page*

*Figure 9 continued*

have diet assigned based on preserved meals (*Eoalulavis* [**Sanz et al., 1996**], *Sapeornis* [**O'Connor, 2019a**]) or quantitative diet proxies (**Miller et al., 2022**; **Miller et al., 2023**). Remaining diets assignments are based on qualitative morphology and depositional setting (**Table 11**). The diet of the common ancestor of Enantiornithes remains obscure, though many late-diverging enantiornithines are recovered as ancestrally invertivorous.

The online version of this article includes the following figure supplement(s) for figure 9:

**Figure supplement 1.** Ancestral state reconstruction of enantiornithine body mass.

**Figure supplement 2.** Ancestral state reconstruction of enantiornithine mechanical advantage (MA) and functional indices.

**Figure supplement 3.** Ancestral state reconstruction of enantiornithine mesh-weighted arithmetic mean (MWAM) strain under bite loading in finite element analysis (FEA).

**Figure supplement 4.** Ancestral state reconstruction of enantiornithine pedal traditional morphometric (TM) variables.

---

The quadrate is disarticulated in every published bohaiornithid skull, and its position is highly influential on MA measurements. Our sensitivity analysis of the quadrate position (**Figure 2—figure supplement 1**, **Supplementary file 4**) indicates that folivory cannot be ruled out for any bohaiornithid. In fact, it is the most likely diet for every taxon except *Parabohaiornis* when the quadrate is shifted to the anteriormost plausible position. Unlike in longipterygids and pengornithids (**Miller et al., 2022**; **Miller et al., 2023**), scavenging is generally not recovered as likely with a posterior quadrate shift except in *Parabohaiornis*, and in this taxon it is also recovered as likely with an anterior shift as well. A posterior-shifted quadrate does, however, make piscivory plausible in *Sulcavis* and *Zhouornis* and the most likely diet for *Shenqiornis*. A posterior quadrate shift also makes invertivory more likely than generalist feeding for *Zhouornis*.

## Species-level interpretations

*Bohaiornis* has unusually high OMA (0.16 lower and 0.33 upper; **Figure 4—figure supplement 2E and F**). Folivores have a diagnostically high OMA ($\bar{x}$ = 0.10 lower, 0.27 upper; **Figure 4—figure supplement 2E and F**), making this the strongest affinity for this taxon (**Table 4**). Notably, though, it has a PMA below any extant folivore (0.16 lower jaw PMA and 0.09 upper jaw PMA, vs a folivore minimum of 0.24 lower jaw PMA and 0.18 upper jaw PMA; **Figure 4—figure supplement 2C and D**). Note also that OMA is strongly affected by the length of the cranium, which may be under different developmental constraints in bohaiornithids (see 'Skull traditional morphometrics'). However, if anything one would expect bohaiornithids to have shorter crania and thus lower OMA than an equivalent crown bird, making a distinctly high OMA even more impactful. *Parabohaiornis* has considerable affinity with swallowing granivores and folivores (**Table 4**), respectfully due to its relatively high AMA and PMA, and lower jaw OMA (**Figure 4—figure supplement 2A–F**).

The remaining bohaiornithids plot within the region of undifferentiated diets (**Figure 4**) and have the strongest affinity with dietary generalists (**Table 4**), previously interpreted as a 'default' prediction in longipterygids (**Miller et al., 2022**). These taxa, however, do show greater affinity with frugivores and less with invertivores and piscivores compared to pengornithids (though *Shenqiornis* has nearly the same affinity with piscivores), bringing all three diets into roughly equal likelihood after generalist feeding. The exception is *Sulcavis*, whose slightly higher OMA increases affinity with folivores rather than frugivores. The posterior skull of *Shenqiornis* is largely reconstructed from cf. *Sulcavis* material (**Figure 2D**), so its membership in this functional guild is less certain and should be revisited as additional material is discovered.

## Evaluating bohaiornithid durophagy

Durophagy, as hypothesised in bohaiornithids (**O'Connor and Chiappe, 2011a**; **O'Connor et al., 2013**; **Chiappe and Meng, 2016**; **Zhou et al., 2021**), is poorly defined. Thus we cannot clearly define a 'durophage' diet category to test this hypothesis. Rather, we can test it indirectly by comparing bohaiornithid jaw performance to that of husking granivores and parrots. Husking granivores are labelled as such in this work based on regular recorded behaviour of cracking hard seeds (**Table 2**), and parrots are the archetypical avian durophages with adaptations across the clade for withstanding

**Table 7.** Quantitative ancestral states of the common ancestor of Enantiornithines.
Mean values are the average of 10,000 random tree permutations where polytomies are randomly resolved. 95% confidence intervals are given as the 2.5 and 97.5% quantiles of the permutations, though upper and lower bounds for all values are identical to seven significant figures. As these values are based on multiple possible topologies of the enantiornithine tree, we consider them more valid than those visualised in figure supplements of *Figure 9*.

| Line of evidence | Trait | Mean | 95% CI low | 95% CI high |
|---|---|---|---|---|
| Mass | Body mass (g) | 198 | 198 | 198 |
| | AMAup | 0.13 | 0.13 | 0.13 |
| | PMAup | 0.19 | 0.19 | 0.19 |
| | OMAup | 0.19 | 0.19 | 0.19 |
| | AOup | 0.09 | 0.09 | 0.09 |
| | MCHup | 0.32 | 0.32 | 0.32 |
| | ACHup | 0.19 | 0.19 | 0.19 |
| | AMAlow | 0.21 | 0.21 | 0.21 |
| | PMAlow | 0.32 | 0.32 | 0.32 |
| | OMAlow | 0.08 | 0.08 | 0.08 |
| | AOlow | 0.065 | 0.065 | 0.065 |
| | MMHlow | 0.12 | 0.12 | 0.12 |
| MA | AMHlow | 0.07 | 0.07 | 0.07 |
| FEA | MWAM strain (µε) | 227 | 227 | 227 |
| | DI/DIII ratio | 0.76 | 0.76 | 0.76 |
| | DII/DIII ratio | 0.85 | 0.85 | 0.85 |
| | DIV/DIII ratio | 0.60 | 0.60 | 0.60 |
| | DI angle (°) | 120 | 120 | 120 |
| | DII angle (°) | 98 | 98 | 98 |
| | DIII angle (°) | 90 | 90 | 90 |
| Pedal TM | DIV angle (°) | 71 | 71 | 71 |

MA = mechanical advantage; FEA = finite element analysis; TM = traditional morphometric; AMA = anterior jaw-closing mechanical advantage; PMA = posterior jaw-closing mechanical advantage; OMA = jaw-opening mechanical advantage; AO = relative articular offset; MCH = relative maximum cranial height; MMH = relative maximum mandible height; ACH = relative average cranial height; AMH = relative average mandible height; MWAM = mesh-weighted arithmetic mean.

high bite forces (*Bright et al., 2019*). The highest jaw-closing mechanical advantage for a bohaiornithid, that of *Parabohaiornis*, is higher than that of the nectarivorous brown lory (*Chalcopsitta duivenbodei*; upper jaw AMA 0.12 in lory vs 0.15 in *Parabohaiornis*), but smaller than every other parrot and husking granivore. The average for upper jaw AMA for parrots (0.23) and husking granivores (0.24) is much higher. The maximum relative jaw height in bohaiornithids is also below the minimum in parrots, and the same is true for husking granivores in the lower jaw. The average relative maximum cranial

**Table 8.** Posterior probability of diet in the common ancestor of Enantiornithes based on mechanical advantage (MA) and functional indices.
Generalist feeding is the most likely, followed by piscivory and invertivory. Diet abbreviations: GranivoreH, Husking Granivore; GranivoreS, Swallowing Granivore; Tetra Hunt, Tetrapod Hunter.

| Folivore | Frugivore | Generalist | GranivoreH | GranivoreS | Invertivore | Nectarivore | Piscivore | Scavenger | Tetra Hunt |
|---|---|---|---|---|---|---|---|---|---|
| 2.53E-03 | 5.58E-03 | 6.51E-01 | 2.17E-07 | 6.44E-03 | 7.57E-02 | 3.99E-03 | 2.12E-01 | 4.28E-02 | 4.41E-04 |

height (MCH) of husking granivores and bohaiornithids is nearly the same (0.35 and 0.34, respectively). In short, the upper jaw in bohaiornithids is somewhat robust, but the upper and lower jaws lack force production adaptations seen in extant avian durophages.

## Finite element analysis

Bohaiornithid jaws are stronger than both longipterygids' and pengornithids'. Under the same construction and loading conditions, longipterygid jaws ranged from 259 to 354 με (*Miller et al., 2022*) and pengornithid jaws from 190 to 275 με (*Miller et al., 2023*). Bohaiornithids, in contrast, have jaw strain ranging from 89 to 156 με (*Figure 5*). While they do separate from these other enantiornithine families, the intermediate strength of their jaws compared to extant birds means they overlap with most diets in multivariate space (*Figure 6*). Thus, dietary diagnosis from FEA is less precise in bohaiornithids than previously studied enantiornithines.

### Species-level interpretations

In lieu of a clear separation in multivariate space, we can still consider diets unlikely for a taxon if the MWAM strain of the jaw under loading falls outside the range of any birds with that diet. However, unlike in Longipterygidae in which the fossil taxa's jaws are weaker than many extant diet groups (*Miller et al., 2022*), bohaiornithid jaws are stronger than most extant birds'. Rather than being unable to process a given food as in a weaker jaw, we interpret stronger jaws as 'overbuilt' for a given diet. This implies the jaw evolved under pressures for more durable food, but does not necessarily prohibit consumption of more compliant foods. *Bohaiornis*, *Shenqiornis*, *Sulcavis*, and *Zhouornis* experience less strain than any swallowing granivore in the extant dataset; *Bohaiornis*, *Shenqiornis*, and *Zhouornis* also experience less than any tetrapod hunter; and *Bohaiornis* and *Zhouornis* experience less than any carnivore (*Figure 5*).

In multivariate space (*Figure 6A*), it is apparent that bohaiornithid jaws tend to have a heterogeneous strain distribution. This contrasts with both scavengers and tetrapod hunters, which have a diagnostically homogeneous strain distribution. To a lesser extent frugivores also tend to have homogeneous strain, though the frugivorous African green pigeon *Treron calva* is notably the nearest neighbour of *Zhouornis*. *Shenqiornis* and *Sulcavis* notably do not plot near one another despite the angular and much of the surangular in the *Shenqiornis* reconstruction coming from cf. *Sulcavis* (*Figure 2D*), owing largely to the more robust posterior dentary in *Shenqiornis*.

### Evaluating bohaiornithid durophagy

Comparison to husking granivores and parrots warrants additional attention to evaluate the hypotheses of durophagy. Bohaiornithids have overall low jaw strain under loading (89–156 με, x̄ = 128 με). This is on par with husking granivores (78–161 με, x̄ = 123 με), but weaker than parrots (57–95 με, x̄ = 78 με). Husking granivores plot across PC2 in *Figure 6*, but parrots and bohaiornithids both experience mainly heterogeneous strain patterns. The locations of high and low strains differ between the two groups, though. Parrots tend to experience high strain at the rostral tip of the jaw, while the posterior end of the jaw is under minimal strain. Conversely bohaiornithids tend to experience high strains along the dorsal edge of the jaw, particularly near the dentary/surangular suture, and low strains along the ventral edge of the jaw. This may be partially explained by the lack of a keratinous beak in bohaiornithids, which acts as a strain sink in the rostral jaw of beaked animals (*Lautenschlager et al., 2013*). Bohaiornithids have jaws of similar overall strength to husking granivores but weaker than parrots, and the patterns of strain they experience during a bite are not analogous to either group.

## Traditional morphometrics

### Pes

It has come to our attention (*Clark et al., 2023*) that in the work our pedal morphometric results were originally reported in *Miller et al., 2022* only one non-raptorial perching taxon has an anisodactyl toe arrangement, as in all known enantiornithines. Most are zygodactyl (Cuculidae, Psittaciformes) or semi-zygodactyl (Musophagidae) (*Botelho et al., 2015*). The category revision in *Miller et al., 2023* increased anisodactyl non-raptorial perching to five taxa, but this still represents only about 25% of the non-raptorial perching taxa in the sample. In response to this, we added an additional

five representatives of non-raptorial perching taxa with anisodactyl toe arrangements (Cracidae, Fregatidae, Meropidae, Phoeniculidae, Podargidae). This increased the anisodactyl percentage of the non-raptorial perching birds to 43%.

The new extant data had minimal effect on ecological category boundaries and phylogenetic signal of pedal morphometrics. The largest difference between these results and those in *Miller et al., 2023* is the non-raptorial perching bird space infiltrating a space of slightly less claw curvature (driven by the blunt unguals of the tawny frogmouth *P. strigoides*) and shrikes being slightly more distinct in pFDA. No categories gain or lose significant differences in phylogenetic HSD from *Miller et al., 2023*. Given the minimal change caused by adding more anisodactyl birds, that the anisodactyl birds plot alongside our zygodactyl non-raptorial perchers, and that owls (Strigidae) and the osprey (Pandionidae) are semi-zygodactyl (*Botelho et al., 2015*) but plot far from zygodactyl non-raptorial perchers, we believe that toe arrangement is not a driving force of the trends observed in our pedal morphometric data. Certain toe arrangements in which digits I and II are not the primary grasping digits, such as heterodactyly, are not included in this study and may have meaningfully different patterns of claw curvature and size.

Bohaiornithid claw curvature and relative size are generally conserved through ontogeny (*Figure 7A*), more like *Bubo virginianus* (Figure 3 in *Fowler et al., 2011*) than *Longipteryx* (Figure 5 in *Miller et al., 2022*). Thus, ontogeny is not considered a major factor in claw TM in this study.

## Evaluating bohaiornithid raptorial behaviour

As previously proposed (*Wang, 2014a*), most bohaiornithid talons plot near some extant raptorial birds (*Figure 7*) and have some affinity with them in FDA and pFDA (*Table 6*). However, most taxa also have some level of non-raptorial affinity. We thus consider raptorial behaviour only likely in certain taxa, discussed below.

## Species-level interpretations

*Longusunguis* is the bohaiornithid most consistently recovered here as raptorial. It plots within or near large raptors in every analysis (*Figure 7*) and is consistently recovered as likely to be a large raptor (*Table 6*). The joints of digits I and II are somewhat hinged (=ginglymoid *sensu*; *Fowler et al., 2011*), though digits III and IV are unhinged (*Hu et al., 2020b*), indicating grasping ability focussed in the first two digits as in many extant accipiters (*Fowler et al., 2009*). As noted by *Wang, 2014a*, the tarsometatarsus of *Longusunguis* and other bohaiornithids is relatively short and robust, typically interpreted as increasing grip strength at the cost of speed (*Fowler et al., 2009*; *Einoder and Richardson, 2007*; *Ward et al., 2002*) and more common among raptorial avians in ambush predators (*Einoder and Richardson, 2007*).

*Sulcavis* has the strongest raptorial affinity in PCA (*Figure 7A*) and has some raptorial affinity in FDA and pFDA (*Figure 7B*, *Table 6*). Unexpected for a raptorial bird, the specimen displays a relatively small and straight digit II ungual. Conversely, digit I is the most enlarged and recurved of any bohaiornithid and among the most enlarged and recurved in the dataset overall. Enlargement of the hallux without enlargement of any opposing digit is uncommon in extant birds, mostly occurring in larks (Alaudidae) in which the digit I is also nearly straight. Digit II is damaged in the holotype of *Sulcavis* (*O'Connor et al., 2013*), and it may be that our estimate of its extent does not reflect its actual enlargement and curvature. If this is the case, then it likely used its feet raptorially as we hypothesise for *Longusunguis*, but if not then this may indicate some unique use for digit I other than raptorial grasping.

*Zhouornis* and *Bohaiornis* can reasonably be interpreted either as small raptors or non-raptorial perchers. Both have strong affinity with small raptors in FDA (*Figure 7B*, *Table 6*) but also plot near ground birds and non-raptorial birds in PCA (*Figure 7A*). Their claw curvature (average 91°) is dissimilar from ground birds, rendering that diagnosis less likely. Their phalanges are moderately hinged (*Li et al., 2014*; *Hu et al., 2011*; *Zhang et al., 2013*; *Zhang et al., 2014*) which indicates some level of grasping adaptation, useful both for raptors and non-raptorial perchers. The more mature specimen of each bird (B2 and Z1 in *Figure 7A*; see *Figure 10* regarding maturity) has claws less similar to raptors and more similar to non-raptorial perchers. Digits I, II, and IV maintain similar proportions across ontogeny within each species, but in both species digit III increases in relative size in the more mature specimens (less confident in *Bohaiornis* as pes preservation is poor in IVPP V17963; *Li et al., 2014*), explaining the shift in affinity towards non-raptorial birds.

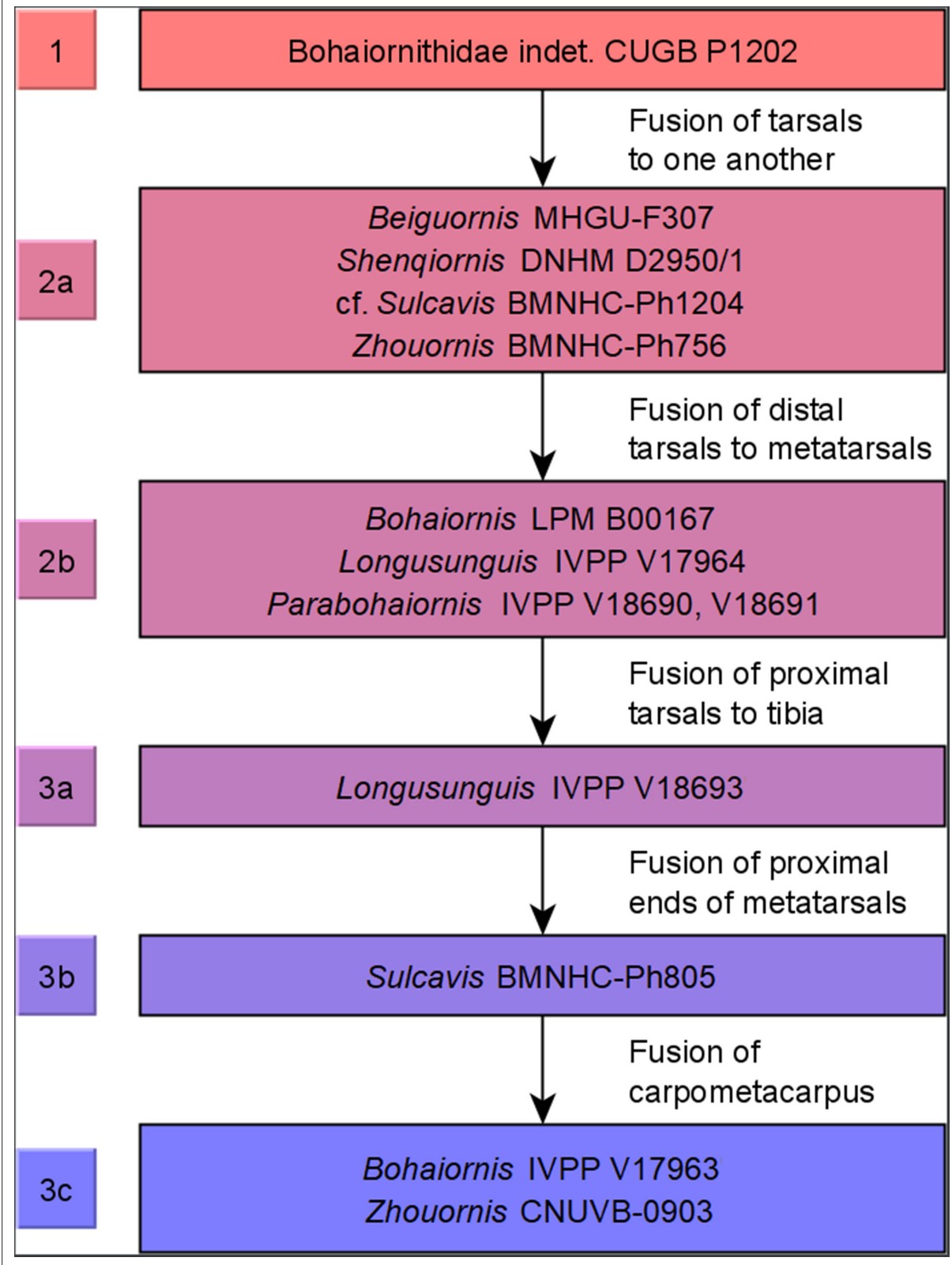

**Figure 10.** Ontogenetic stages of bohaiornithids. Stages are based on *Hu and O'Connor, 2017*, with new subdivisions of stage 3 (possibly specific to Bohaiornithidae) as noted in 'Methods'. Subadult status is reached at or before stage 1 and adulthood within stage 3; see *Table 10* for details.

*Parabohaiornis* consistently resolves as a non-raptorial perching bird across all analyses (*Figure 7*, *Table 6*). The taxon has a relatively small digit I compared to other bohaiornithids, driving this finding. Published image resolution is too poor to tell how hinged their phalanges are (*Wang, 2014a*).

## Skull

### Extant birds

Extant results of our skull TM analysis generally resemble those presented by *Clark et al., 2023*. Our *Figure 8—figure supplement 1A*, compared to their Figure 4, identifies additional separation within herbivorous birds with granivores and folivores separating along the LOG skull length axis. Combining the insectivore and (non-insectivorous) invertivore classes of *Clark et al., 2023* had little effect on the data, as the latter was completely within the former's region of the morphospace in the original dataset. We additionally plotted the ratio of rostrum length to skull length against LOG body mass to investigate if the separation along the skull length axis was a size effect, and the results in *Figure 8— figure supplement 1B* support this hypothesis. The structure of the data is overall similar with a slightly more normal spread along the x-axis, but group separation along the x-axis is maintained and overall follows the larger trends in body mass presented in this work.

As size trends in diet are already discussed in this work, we also investigated the relative length of the skull as an alternative variable to plot against the relative rostral length (*Figure 8*). Nectarivorous taxa largely display very long skulls relative to body size, though this is driven entirely by humming-birds (Trochilidae). Hummingbirds are known to have skulls that are peramorphic relative to other Strisores (*Navalón et al., 2021*), possibly to support requisite neural tissue despite miniaturisation (*Ocampo et al., 2018*). Aside from this group, relative rostrum length and relative skull length share a general positive correlation. In other words, elongation of the rostrum tends to parallel elongation of the skull. This is unsurprising as the rostrum in extant birds has relatively high integration with the rest of the skull (Tables S2 and S3 in *Felice and Goswami, 2018*). General relationships between skull and rostrum length and diet parallel those we discussed previously (*Miller et al., 2022*) for AMA and OMA. Short-skulled birds (e.g. granivores) have relatively sturdy jaws more efficient at processing hard and tough foods, while those with longer jaws (e.g. piscivores) can more efficiently move their jaws at high speeds and thus better catch quick-moving prey.

### Fossil birds

Skull traditional morphometrics, while useful in determining longipterygid diet (*Clark et al., 2023*), appear less useful when examining bohaiornithids. Bohaiornithids have rostra to skull length ratios clustered around 0.8, a region inhabited by birds of all diets except nectarivores and granivores. Bohaiornithids also do not conform to the general correlation of increasing rostrum length and skull length. If anything, those with longer skulls have shorter rostra.

We offer two possible interpretations. Firstly, some bohaiornithids may have skull functions not discussed here. *Bohaiornis*, *Longusunguis*, and *Parabohaiornis* do plot at the edge of the extant space near folivores (*Opisthocomus hoazin* and two phasianids), frugivores (two psittacoids), and a generalist (*Numida meleagris*). *Shenqiornis*, *Sulcavis*, and *Zhouornis*, however, plot outside the extant space and may represent an ecology dissimilar from any sampled extant bird.

Alternatively, this may indicate different developmental constraints on the skull in bohaiornithids (and possibly enantiornithines as a whole). One hypothesis proposes that crown birds ancestrally have relatively larger brains than their earlier-diverging counterparts (*Torres et al., 2021*), which may necessitate relatively longer skulls to house the larger brain. However, a subsequently discovered enantiornithine basicranium displays many crown-bird-like characteristics and has been used to argue a crown-bird-like endocranium was present in the common ancestor of Ornithothoraces (*Chiappe et al., 2022*). Given this uncertainty, we use skull TM only tentatively to interpret ecology in bohaior-nithid taxa and generally recommend caution in interpreting non-crown bird ecology from measures including the cranium.

## Bohaiornithid ecology and evolution

By combining the individual lines of evidence discussed above, more precise diet diagnoses can be obtained. *Table 9* summarises these diagnoses, with elaboration by species below.

**Table 9.** Summary table of interpretations of each line of evidence used herein. Body mass, mechanical advantage (MA), and finite element analysis (FEA) inform diet. Pedal traditional morphometric (TM) informs the use of the pes or lack thereof in feeding. See relevant discussion sections for additional details. Bolded diets are agreed upon by all available diet proxies. Bolded pedal ecologies are either help discriminate diet (carnivory in *Longusunguis*) or are supported over other possibilities by diet information (non-raptorial perching in *Bohaiornis*). Given the uncertain application of skull TM to bohaiornithids, diets from this proxy are not bolded.

| Line of evidence | Taxon | Likely diets/ecologies | Unlikely diets/ecologies |
|---|---|---|---|
| | *Beiguornis* | Folivore, Frugivore, Generalist, Piscivore, Tetrapod Hunter | Granivore, Invertivore, Nectarivore |
| | *Bohaiornis* | **Folivore**, Frugivore, Generalist, Piscivore, Tetrapod Hunter | Granivore, Invertivore, Nectarivore |
| | *Longusunguis* | Folivore, Frugivore, **Generalist**, **Piscivore**, Tetrapod Hunter | Granivore, Invertivore, Nectarivore |
| | *Parabohaiornis* | **Folivore**, Frugivore, **Generalist**, Piscivore, Tetrapod Hunter | Granivore, Invertivore, Nectarivore |
| | *Shenqiornis* | **Folivore**, Frugivore, **Generalist**, **Piscivore**, Tetrapod Hunter | Granivore, Invertivore, Nectarivore |
| | *Sulcavis* | **Folivore**, Frugivore, **Generalist**, **Piscivore**, Tetrapod Hunter | Granivore, Invertivore, Nectarivore |
| | *Zhouornis* | Folivore, **Frugivore**, **Generalist**, Piscivore, Scavenger, Tetrapod Hunter | Granivore, Invertivore, Nectarivore |
| Body mass | *Bohaiornis* | **Folivore**, Generalist, Swallowing Granivore | Husking Granivore, Scavenger |
| | *Longusunguis* | Frugivore, **Generalist**, Invertivore, **Piscivore** | Husking Granivore, Scavenger |
| | *Parabohaiornis* | **Folivore**, **Generalist**, Scavenger, Swallowing Granivore | Husking Granivore, Tetrapod Hunter |
| | *Shenqiornis* | **Folivore**, Frugivore, **Generalist**, Invertivore, **Piscivore** | Husking Granivore, Tetrapod Hunter |
| | *Sulcavis* | **Folivore**, **Generalist**, Invertivore, **Piscivore** | Husking Granivore, Scavenger |
| | *Zhouornis* | Frugivore, **Generalist**, Invertivore, Piscivore | Husking Granivore, Scavenger |
| Mechanical advantage | | | |

*Table 9 continued on next page*

*Table 9 continued*

| Line of evidence | Taxon | Likely diets/ecologies | Unlikely diets/ecologies |
|---|---|---|---|
| Finite element analysis | *Bohaiornis* | **Folivore**, Husking Granivore, Nectarivore | Swallowing Granivore, Invertivore, Piscivore, Scavenger, Tetrapod Hunter |
| | *Longusunguis* | Folivore, Frugivore, **Generalist**, Husking Granivore, Swallowing Granivore, Invertivore, Nectarivore, **Piscivore** | Scavenger, Tetrapod Hunter |
| | *Parabohaiornis* | **Folivore**, Frugivore, **Generalist**, Husking Granivore, Swallowing Granivore, Invertivore, Nectarivore, Piscivore | Scavenger, Tetrapod Hunter |
| | *Shenqiornis* | **Folivore**, Frugivore, **Generalist**, Husking Granivore, Invertivore, Nectarivore, **Piscivore** | Swallowing Granivore, Scavenger, Tetrapod Hunter |
| | *Sulcavis* | **Folivore**, Frugivore, **Generalist**, Husking Granivore, Invertivore, Nectarivore, **Piscivore** | Swallowing Granivore, Scavenger, Tetrapod Hunter |
| | *Zhouornis* | **Frugivore, Generalist**, Husking Granivore, Nectarivore | Swallowing Granivore, Invertivore, Piscivore, Scavenger, Tetrapod Hunter |
| Pedal traditional morphometrics | *Bohaiornis* | **Non-Raptorial Perching**, Small Raptor | Ground Bird, Shrike-like |
| | *Longusunguis* | **Large Raptor** | Ground Bird, Shrike-like |
| | *Parabohaiornis* | Non-Raptorial Perching | Ground Bird |
| | *Sulcavis* | Large Raptor, unique ecology not in dataset | Shrike-like |
| | *Zhouornis* | Small Raptor, Non-Raptorial Perching | Ground Bird, Shrike-like |
| Skull traditional morphometrics | *Bohaiornis* | Folivore, Frugivore, Generalist | Granivore, Nectarivore |
| | *Longusunguis* | Folivore, Frugivore, Generalist | Granivore, Nectarivore |
| | *Parabohaiornis* | Folivore, Frugivore, Generalist | Granivore, Nectarivore |
| | *Shenqiornis* | Unique ecology not in dataset | All sampled diets |
| | *Sulcavis* | Unique ecology not in dataset | All sampled diets |
| | *Zhouornis* | Unique ecology not in dataset | All sampled diets |

## Species-level diet diagnoses

*Bohaiornis* has jaws highly reminiscent of avian folivores (note that we use this term to refer to animals primarily consuming any non-reproductive plant tissues, not strictly leaves; *Figure 1*, centre). Its relative skull and rostrum size is highly similar to folivores, and its body mass falls within the folivore range as well. As noted in the MA discussion, the most uniquely folivorous trait of this taxon is a high OMA. While the PMA of *Bohaiornis* is below that of any studied folivore, its jaw strength in FEA is also similar to that of folivores. The other groups it plots near in FEA function space (*Figure 6A*), husking granivores and nectarivores, tend to be much less massive than *Bohaiornis*. A herbivorous diet would imply the claws were not used to kill prey, which is plausible for *Bohaiornis* (*Figure 7*). Perching/arboreal lifestyles for living folivorous birds are uncommon, and these birds are typically weak fliers (*Dudley and Vermeij, 1992*). The hoatzin (*O. hoazin*) and southern screamer (*Chauna torquata*) have both similar OMA (respectively 0.37 upper 0.07 lower and 0.28 upper 0.13 lower) and MWAM strain (both 121 με) to *Bohaiornis* (upper OMA 0.33, lower OMA 0.16, MWAM strain 104 με). Both of these extant folivores typically fly only short distances with continuous flapping (*Billerman, 2020*; *Brady, 2020*) which is also the flight style predicted for bohaiornithids (*Chiappe et al., 2019*). These two taxa serve as the most likely extant analogues for *Bohaiornis*; either birds which spent most of their lives feeding and climbing in trees like the hoatzin (*Billerman, 2020*) or more ground-based foragers who use trees as a refuge when resting or threatened (*Brady, 2020*). *Bohaiornis* IVPP 17963 plots near the hoatzin in pedal TM (*Figure 7A*), so the former lifestyle seems more likely for at least a mature *Bohaiornis*.

*Parabohaiornis* has strong herbivorous affinities, though they are less clear than those in *Bohaiornis*. Its skull is most similar to swallowing granivores in both MA (*Table 4*) and FEA (*Table 5*), though it is more massive than most extant granivorous birds (178–406 g vs upper cut-off of 53 g). Sandgrouse (Pteroclidae) may serve as an extant analogy as swallowing granivores in the 200–300 g mass range (*Winkler et al., 2020*). The two sandgrouse in this study, *Pterocles exustus* and *Pterocles orientalis*, have overall similar MA (e.g. upper jaw AMA 0.18 and 0.19, 0.15 in *Parabohaiornis*) and skull robusticity (ACH 0.24 and 0.23, 0.25 in *Parabohaiornis*), but their jaws are weaker under loading in FEA (MWAM strain 311 and 273 με, 156 με in *Parabohaiornis*). Also of note, sandgrouse are generally terrestrial while the unguals of *Parabohaiornis* indicate an arboreal lifestyle. Alternatively, *Parabohaiornis* shows the next most affinity with folivores, and this diagnosis is more consistent with its large body mass and relatively short skull and intermediate-length rostrum. Like *Bohaiornis*, *Parabohaiornis* is adapted for non-raptorial perching and would likely be analogous to a hoatzin if folivorous. Generalist feeding is recovered as possible for Parabohaiornis in MA and FEA, but consistently less so than granivory and folivory so it is treated here as a 'default prediction' (*Miller et al., 2022*) and not discussed at length.

The crop and gizzard (i.e. ventriculus or gastric mill) play major roles in folivory and granivory for extant avians, serving as a site to process the tough plant matter during digestion (*Dudley and Vermeij, 1992*; *Gionfriddo and Best, 1999*). Crops and gizzards have previously been inferred to be absent in Enantiornithes (*O'Connor and Zhou, 2020c*), which could potentially be strong evidence against folivory or granivory in *Bohaiornis* and *Parabohaiornis*. Two (non-exclusive) possibilities may still allow for effective herbivory in these taxa: oral processing and hindgut specialisation. Most groups of herbivorous lizards have minimal postcranial digestive specialisation, and instead are able to orally process plants to a level where nutrients can be efficiently extracted (*Cooper and Vitt, 2002*). The lack of any accessory cusps as seen in many herbivorous lizard clades (*Melstrom, 2017*) is unsurprising in bohaiornithids, given the simplification of enamel at the Avialae node (*Li et al., 2020*). Better analogues are skinks (Scincidae) and tegus (Teiidae), which retain conical dentition which is simply more robust than their carnivorous and omnivorous relatives (*Melstrom, 2017*). Bohaiornithids, notably, have some of the most robust teeth within Enantiornithes (*O'Connor and Chiappe, 2011a*). This may have been an adaptation or exaptation for processing plant matter. Additionally, several groups of extant folivorous birds have specialised hindguts to process plant matter. Both emus (*Herd and Dawson, 1984*) and geese (*Buchsbaum et al., 1986*) have regions of their digestive tract specialised in fermenting fibrous foods, and anseriform (*Ankney and Afton, 1988*; *Kehoe et al., 1988*) and galliform (*Gasaway, 1976*; *Moss, 1989*; *Remington, 1989*) birds are known to increase their intestinal lengths when consuming high-fibre diets, which allows for a higher percentage to be digested. While we lack any direct evidence of these adaptations in Bohaiornithidae, we contest there is evidence for specialised hindgut digestion of plant material in enantiornithines more broadly. Structures now recognised as plant propagules (*Mayr et al., 2020*) have been identified in seven enantiornithine

specimens (*O'Connor et al., 2014*). Of these, four specimens are figured (STM 10-45, STM 29-8 [*Zheng et al., 2013*], STM 10-12 [*O'Connor et al., 2014*], and STM 11-80 [*Wang et al., 2016b*]), and in these figured specimens the propagules are preserved either in the lowest thoracic (STM 10-45 and STM 11-80) or pelvic (STM 29-8 and STM 10-12) region. As previously noted, consumulite preservation is most likely with long gut retention times (*Miller and Pittman, 2021*), so preservation of these plant propagules specifically in the region inhabited by the hindgut in life implies a prolonged retention in this area where plant matter could ferment and break down. So, despite likely lacking a crop and gizzard, *Bohaiornis* and *Parabohaiornis* could have sustained folivory with oral processing of plant matter and/or specialised fermentation in the hindgut.

Alternatively, as noted in the FEA discussion, our diagnosis of *Bohaiornis* and *Parabohaiornis* as herbivorous is based on the relatively high strength and efficiency of their jaws. These adaptations indicate evolutionary pressure for the ability to consume plant material, but not necessarily a regular habit of doing so. 'Fallback foods', foods avoided during normal feeding but essential to survival when alternatives are unobtainable, are well-documented evolutionary drivers of extant animals' functional morphology (*Marshall and Wrangham, 2007*). The Jehol climate is believed to have been highly variable and affected by local volcanic activity (*Xu et al., 2020*), so specialised herbivory may have been restricted to times when this instability limited other food resources. If this is the case, then specialised oral or hindgut processing may not have been necessary to allow *Bohaiornis* and *Parabohaiornis* to survive short-term resource scarcity.

*Zhouornis* has the strongest jaw among bohaiornithids (*Figure 5*), with total evidence pointing to either a frugivorous or generalist lifestyle. Only frugivores, generalists, husking granivores, and nectarivores inhabit a region of the FEA strain-space where jaw strain is as low and heterogeneous as in *Zhouornis*. This space is generally sparsely occupied and thus seems ecologically meaningful. However, *Zhouornis* is also the largest bohaiornithid (mean estimate 758 g for the largest specimen), and very few extant granivorous or nectarivorous birds reach this size (*Figure 3*). Frugivory and generalist feeding are also recovered as likely from MA posterior predictions (*Table 4*), so these two diets are considered most likely for *Zhouornis*. Pedal TM results could support either diet, with talons similar to both raptors taking small prey (potentially useful for a generalist) and non-raptorial perching birds (as expected of a frugivore). If *Zhouornis* was indeed a generalist, it would potentially have an interesting niche overlap with that proposed for *Pengornis* (*Miller et al., 2023*). *Pengornis* shows signs of generalist feeding with ability to raptorially take large prey. While *Zhouornis*' pes more resembles that of raptors taking prey that is small relative to body size, *Zhouornis*' absolute body size is much larger than *Pengornis* (mean mass estimate 437 g; *Serrano et al., 2015*). This may represent a different approach to taking advantage of the same prey resources. We predict *Zhouornis* would be capable of macrocarnivory but less suited for it than *Pengornis*; the barred owl (*Strix varia*) has a similar body mass and claw shape to *Zhouornis*, and while it is capable of taking large prey like grouse and rabbits it preferentially takes mouse-sized prey (*Mazur and James, 2021*).

*Longusunguis*, *Shenqiornis*, and *Sulcavis* have less clear diagnoses. Piscivory and generalist feeding are recovered as likely for all three, with folivory also likely for *Shenqiornis* and *Sulcavis*. Unlike *Parabohaiornis*, however, there is no clear hierarchy of these possibilities. Much of the uncertainty stems from FEA giving little useful data, with these taxa having intermediate MWAM strains when loaded (*Figure 5*) and inhabiting a region of the strain-space shared by most diets (*Figure 6A*) only ruling out scavenging and tetrapod hunting. As mentioned above, their intermediate mass can also only rule out granivory, invertivory, and nectarivory. We are therefore left with MA to discriminate between the remaining diets, which has poor predictive power in extant birds. Generalist feeding and piscivory are recovered as somewhat likely for all three from MA (*Table 4*), though as previously proposed this could represent a default prediction (*Miller et al., 2022*). Folivory can also not be ruled out in any of these taxa due to uncertainty in quadrate placement (*Supplementary file 4*). Pedal TM, however, points to folivory being unlikely in *Longusunguis* as its talons appear adapted for taking large animal prey. *Longusunguis* notably has more recurved teeth than other bohaiornithids (Figures 4 and 6 in *Zhou et al., 2021*) which is associated with carnivory (*Button and Zanno, 2020*). Skull TM of *Longusunguis* are most similar to folivorous birds, but given the uncertainty in what developmental constraints are at play we consider folivory overall less likely in *Longusunguis* than carnivory. Claw shape may also point to carnivory in *Sulcavis*, though as discussed above its raptorial affinities are questionable. *Shenqiornis* only preserves two disarticulated talons (*Wang et al., 2010*), and while these superficially resemble

the claws of *Zhouornis* in size and curvature we cannot quantitatively reject hypotheses of the pes not being adapted for prey capture. In turn, we cannot reject hypotheses of folivory in *Shenqiornis* or *Sulcavis* (*contra* our previous carnivorous assignment of *Shenqiornis*; *Miller and Pittman, 2021*). Diagnosis of *Shenqiornis* is further complicated by the reliance on cf. *Sulcavis* material to reconstruct its skull (*Figure 2D*), artificially inflating the functional similarity of the two genera. Discovery of additional *Shenqiornis* specimens should clarify the accuracy of this model. In any case, even with four lines of evidence the dietary diagnosis of these three taxa remains tentative, and investigating additional lines of evidence (e.g. dental microwear, muscle reconstruction, or stable isotope analysis; *Miller and Pittman, 2021*) is necessary.

*Beiguornis* preserves neither a skull nor talons (*Wang, 2022a*), so only mass data is available. As with most bohaiornithids, the only diets which can be considered unlikely based on its mass are invertivory, granivory, and nectarivory. Further ecological information will require new techniques or more complete specimens.

## Evaluating bohaiornithid ecological hypotheses

Our results support hypotheses of raptorial behaviour (*Miller and Pittman, 2021*; *Wang, 2014a*; *Chiappe and Meng, 2016*; *Li et al., 2014*) in bohaiornithids, though not for every taxon. We interpret *Longusunguis* as a small predator adapted to take prey too large to be completely encircled by the pes, and *Zhouornis* as potentially a large predator whose talons were adapted to take prey which could be fully encircled by the pes. *Sulcavis* has a greatly enlarged and recurved digit I as expected in a raptor, but lacks any enlarged opposed digit that could be used to grip. As noted above, this could be taphonomy obscuring the full extent of digit II or a different, unaccounted for use of the pes. The talons of *Parabohaiornis* are most akin to non-raptorial perching birds, and while the talons of *Bohaiornis* are between those of raptors and non-raptors we interpret them as non-raptorial given the taxon's herbivorous skull adaptations. Note that the alleged rangle (raptorial gastroliths) used as previous evidence of raptorial behaviour in *Bohaiornis* (*Li et al., 2014*) has been reinterpreted as postmortem mineral growth (*Liu et al., 2021*). Specimens of *Shenqiornis* preserving talons are needed to opine on whether it was adapted for raptorial behaviour. We previously interpreted *Shenqiornis* as carnivorous from a combination of its MA and prior interpretations of all other bohaiornithids as raptorial (*Miller and Pittman, 2021*), but with the variety of pedal ecologies identified in this study we now consider its diet indeterminate.

We also support hypotheses of durophagy (*O'Connor and Chiappe, 2011a*; *O'Connor et al., 2013*; *Chiappe and Meng, 2016*; *Zhou et al., 2021*) in bohaiornithids, though with the caveat that their adaptations are less extreme than what one usually considers an avian durophage. The first proposal of durophagy in a bohaiornithid is made in contrast to proposing most other enantiornithines as 'general insectivore[s], limited to small arthropods with softer cuticles' (*O'Connor and Chiappe, 2011a*). We support Bohaiornithids as durophagous in the sense that they have higher jaw strengths than other modelled enantiornithines (see above). They appear adapted for taking foods harder and tougher than soft-bodied arthropods, but the same could be said for pengornithids. In living birds, 'durophagy' typically refers to cracking hard foods like seeds or nuts within the beak (*Soons et al., 2010*). As examined in the MA and FEA discussion sections, bohaiornithids resemble extant husking granivores in both the relative height of their upper jaw and the overall strain under loading of their lower jaw; however, they lack the adaptations for bite force production of husking granivores and fall short of parrots in all potential durophagy metrics. Bohaiornithids, therefore, do not display adaptations for taking foods as hard as extant birds commonly considered durophages.

Based on our findings herein, we encourage caution in future works using hypothesised diet as a variable. Notably when (*Zhou et al., 2021*) investigated avialan tooth shape in relation to diet, bohaiornithids are placed into the single 'durophage' diet category and are its only members. Bohaiornithid tooth shape does tightly cluster in the study's phylomorphospaces (Figures 4–6 in *Zhou et al., 2021*), but they notably tend to take a central position in the morphospace and overlap with purported granivores, insectivores, and piscivores. Given this overlap and the ecological diversity within the group we support here, we suggest the similarity in bohaiornithid tooth shape is driven by phylogeny rather than, as proposed by *Zhou et al., 2021*, diet. The robust teeth in Bohaiornithidae may serve as general-use tools capable of processing diverse foods effectively rather than adaptations to a specialised diet shared across the clade.

## Evolutionary history of enantiornithine ecology

### Trends within the three major enantiornithine families

Our data suggest that bohaiornithids cover a greater ecological breadth than previously studied enantiornithine families (*Figure 9*). Longipterygids are reconstructed as mainly invertivorous with variability in foraging environment (*Miller et al., 2022*; *Figure 1*, left; though our mass results here suggest the hypothesis of *Longipteryx* as a piscivore cannot be rejected, *contra Miller et al., 2022*, see Body Mass Discussion), and pengornithids are reconstructed as generally piscivorous with *Pengornis* diversifying into a more generalist and macrocarnivorous diet (*Miller et al., 2023*; *Figure 1*, right). In contrast, *Bohaiornis* and *Parabohaiornis* show adaptations reminiscent of extant herbivores while *Longusunguis* and possibly *Zhouornis* were adapted for taking vertebrate prey. The differences in the mass and talon shape of the latter pair further imply partitioning of the carnivorous niche.

The increased dietary breadth in Bohaiornithidae appears to stem from increased strength in the bohaiornithid skull. In particular, Longipterygidae and Bohaiornithidae may represent opposite evolutionary trajectories from the skull architecture of Pengornithidae, which is considered plesiomorphic for enantiornithines (*Wang et al., 2021b*), and from the ancestral enantiornithine skull (*Table 7*). Past works (*Wang, 2014a*; *Chiappe et al., 2019*) have qualitatively described bohaiornithid skulls and teeth as robust, though when using ACH to quantify robusticity bohaiornithids are not particularly remarkable. While there is a large difference between longipterygids (ACH 0.09–0.11, $\bar{x}$ = 0.10; *Miller et al., 2022*) and pengornithids (ACH 0.22–0.23, $\bar{x}$ = 0.23; *Miller et al., 2023*), bohaiornithid skulls (ACH 0.21–0.26, $\bar{x}$ = 0.24) are only slightly more robust than those of pengornithids. Ancestral state reconstruction (*Figure 9—figure supplement 3K*) predicts Bohaiornithidae and Pengornithidae converged upon this level of robusticity from an ancestor whose ACH was between them and Longipterygidae. What we find more noteworthy than robusticity is a distinct stepwise increase in jaw strength between Jehol enantiornithine families: longipterygids have the weakest jaws (259–345 µε, $\bar{x}$ = 309 µε; *Miller et al., 2022*), with pengornithid jaws stronger (190–275 µε, $\bar{x}$ = 222 µε; *Miller et al., 2023*), and bohaiornithids' stronger still (89–156 µε, $\bar{x}$ = 128 µε). Ancestral state reconstruction supports bohaiornithids and longipterygids diversifying from a common ancestor with pengornithid-like jaw strength (*Figure 9—figure supplement 3*), though a lack of jaw strength data from families outside these three limits the reliability of this reconstruction. An increase in jaw strength may have allowed Bohaiornithidae to process a wider variety of difficult-to-process foods than contemporary groups and ecologically diversify, but the poor resolution of relationships within and near the clade (*Wang et al., 2021b*; *Li et al., 2022*; *Table 1*) makes this possibility difficult to test. For instance, the skull of *Fortunguavis*, while overall poorly preserved, does appear to have a robust surangular and pterygoid (*Wang et al., 2014b*) and is commonly recovered as sister to Bohaiornithidae. Strengthening of the skull may therefore be present in more 'bohaiornithid-like' enantiornithines (*sensu Wang et al., 2022b*) and even convergent between them. *Shenqiornis*, the geologically oldest bohaiornithid, has a jaw strength intermediate within bohaiornithids (*Figure 5*) which does imply the skull strengthening adaptations in Bohaiornithidae at least predate 124 Ma. Longipterygids, conversely, were limited by their weak and gracile jaws to relatively soft and compliant food sources (*Miller et al., 2022*). However, whether the gracile jaws of longipterygids were adaptations to specialise in soft and compliant prey, or had unrelated benefits and coincidentally constrained their diet, remains unclear.

Bohaiornithids and Pengornithids are notably larger than other Jehol enantiornithines (*Miller and Pittman, 2021*). While we interpret only *Longusunguis* and possibly *Zhouornis* as inhabiting similar niches to pengornithids, this size increase does make invertivory unlikely in every pengornithid and bohaiornithid. The food web of the Jehol Biota is generally understood to be dominated by small animals (*Matsukawa et al., 2014*; *O'Connor et al., 2019b*), particularly in the Jiufotang formation in which bohaiornithids and pengornithids are most diverse. Accordingly, one would expect interspecific competition for invertebrates to be high in this ecosystem. Bohaiornithidae and Pengornithidae (alongside the ornithurine family Songlingornithidae; *Miller and Pittman, 2021*), then, may be lineages which attempted to alleviate invertivorous competition by diversifying into consuming vertebrate or plant prey. Alternatively, size increase is also commonly interpreted as defence against predation. *Microraptor* was definitely capable of preying on smaller enantiornithines (*O'Connor et al., 2011b*) and the Jehol Biota had many carnivores larger than *Microraptor* (*Matsukawa et al., 2014*; *O'Connor et al., 2019b*). These possibilities need not be mutually exclusive: the more abundant large carnivores in the Yixian formation (*Matsukawa et al.,*

2014) may have provided pressure to grow larger when these clades first emerged, and the subsequent reduction in large carnivores by the Jiufotang formation (*Matsukawa et al., 2014*) may have allowed them to further diversify. Indeed, ancestral state reconstruction (*Figure 9—figure supplement 1*) predicts a concurrent increase in body mass for Bohaiornithidae and Pengornithidae near 125 Ma. Interestingly, though, it recovers the large mass of pengornithids as convergent, not present ancestrally in the clade. This is driven by the earliest pengornithid, *Eopengornis*, being smaller than its more recent relatives. No bohaiornithid is known from this time frame, so we cannot yet comment on if their increased body mass is similarly convergent or ancestral to Bohaiornithidae as a whole.

## The common ancestor of Enantiornithes and the speed of ecological diversification

Given the above trends, we can begin to speculate on the ancestral enantiornithine diet. The common ancestor of Enantiornithes likely had a body mass near 200 g (*Figure 9—figure supplement 1*, *Table 7*), pointing to frugivory or generalist feeding. Its MA and jaw robusticity was likely intermediate to low (*Figure 9—figure supplement 2*), overall resembling extant generalists, piscivores, and invertivores (*Table 8*). Its jaw strain during a bite is estimated at 227 με (*Table 7*), within the range of Pengornithidae (*Figure 9—figure supplement 3*) and thus consistent with folivory, swallowing granivory, generalist feeding, and invertivory (*Miller et al., 2023*). The claws of the common ancestor of Enantiornithes are shaped in a way that is dissimilar from any extant bird included in this study, plotting between non-raptorial perchers and shrikes (*Figure 7—figure supplement 2*). This is reminiscent of *Eopengornis*, and this claw shape has been interpreted as arboreal and allowing for very limited prey manipulation (*Miller et al., 2023*). Overall, these lines of evidence point to the common ancestor of Enantiornithes being an arboreal generalist feeder. If this is the case, then the numerous ecological specialisations seen within Enantiornithes likely represent lineages already exploiting a resource further specialising in it, rather than dramatic trophic shifts. One enantiornithine lineage, that leading to both Longipterygidae and *Eoalulavis*, does appear to be ancestrally invertivorous (*Figure 9*). Thus the purportedly vertivorous avisaurids (*Chiappe, 1993*; *Chiappe and Calvo, 1994*), late-diverging members of this lineage, would indeed represent a distinct increase in trophic level. However, as illustrated in *Figure 9*, the diet of many of the earliest-diverging enantiornithines remains obscure, and these taxa strongly affect the interpretation of the ancestral state. Our quantitative reconstruction of the common ancestor of Enantiornithes in many ways resembles a smaller pengornithid, but that is to be expected when this family is much closer to the root than other taxa included in MA, FEA, and pedal TM reconstructions. This highlights the importance of further ecological study of the earliest-diverging enantiornithine taxa.

While the ancestral enantiornithine diet remains somewhat speculative (*Figure 9*), it is clear that much ecological change and diversification occurred within Enantiornithes by 120 Ma. If current estimates of the Ornithothoraces node at 145 Ma (*Cau, 2018*; *Wang et al., 2021a*) are correct, then this level of ecological diversity took only 25 million years to achieve. For comparison, Neoaves (containing most trophic diversity in crown birds; *Felice et al., 2019*) had only just diverged 25 million years after the origin of crown birds (*Kuhl et al., 2021*). This suggests, then, that adaptations within crown Aves like edentulism (*Zhou et al., 2021*; *Wang, 2020a*) or cranial kinesis (*Bhullar et al., 2016*) cannot fully explain the rate or extent of ecological diversification of crown birds after the K-Pg extinction (in agreement with several recent studies; *Torres et al., 2021*; *Brocklehurst and Field, 2021*). In fact, crown birds may have been slower to occupy the same ecological breadth as their Mesozoic counterparts. This reframes some questions in evolutionary biology: while the question of what made crown birds uniquely fit to survive the K-Pg extinction (*Torres et al., 2021*; *Bhullar et al., 2016*; *Field et al., 2018*) is still relevant, to find the traits which allowed for their ecological success and diversity we should look at Ornithothoraces as a whole. For instance, Ornithothoraces is defined by a variety of adaptations for improving flight capabilities (*Mayr, 2016*). Flight allows an organism easy access to multiple vertical strata within an environment, and flighted vertebrates tend to have larger home ranges than terrestrial vertebrates (*Flaherty et al., 2010*; *Dodge et al., 2014*). Both of these factors provide volant animals with a greater variety and amount of resources, and may be a driver of repeated rapid diversification in ornithothoracine lineages.

## Methods

### Materials availability checklist

All specimens used herein are held in publicly accessible repositories, as detailed in their relevant methods section. Newly generated data including models and computer code, as well as lists of specimen collection numbers, can be accessed from Mendeley Data (https://doi.org/10.17632/7xtpbv27zh.3).

### Taxonomic reference

We refer to extant taxa based on their genus and species in the Birds of the World database for consistency (*Billerman, 2023*). Within data files, taxa are referred to by their name in the data source (Skullsite Bird Skull Collection [*Van and Jansen, 2020*] or museum specimen designation). Comments in data files note where these identifications differ from Birds of the World or the bird diet database EltonTraits 1.0 (*Wilman et al., 2014*). Designations and relationships of fossil clades are based on *Pittman, 2020b*, though use of 'Bohaiornithidae' in this work is necessarily imprecise due to the instability of the clade (*Table 1*, *Liu et al., 2022*). For reasons explained below, we used Bohaiornithidae to refer to the six taxa referred when the clade was originally defined (*Wang, 2014a*) plus *Beiguornis* (*Wang, 2022a*).

Taxa included in Bohaiornithidae

The taxa included within Bohaiornithidae vary between studies. A summary of taxa included in the group in multiple phylogenies is provided in *Table 1*. The original definition of Bohaiornithidae is "the most recent common ancestor of *Shenqiornis mengi* and *Bohaiornis guoi*, and all its descendants" (*Wang, 2014a*), but in practice this definition is used less strictly. Several studies (*Wang, 2022a*; Figure 3B in *Pittman, 2020b*; *Wang et al., 2015c*; *Wang and Liu, 2016a*) place the Bohaiornithidae node at an earlier diverging position than the strict clade definition in order to include *Zhouornis*, and others (*Chiappe et al., 2019*; Figure 5a in *Liu et al., 2022*; *Li et al., 2022*; *O'Connor et al., 2020b*) recover *Shenqiornis* as so early diverging that most or all enantiornithines would be considered bohaiornithids by the strict definition. *Liu et al., 2022* call for a redefinition of the clade after these findings, but do not provide a formal one. They place the Bohaiornithidae node on its phylogeny with a set of synapomorphies, but include *Gretcheniao* and *Junornis* in the clade which is not supported by subsequent studies (*Wang et al., 2022b*; *Wang, 2022a*; *Wang et al., 2021b*). Overall, the six original taxa proposed as members of Bohaiornithidae by *Wang, 2014a* tend to be recovered as closely related to *Bohaiornis. Longusunguis* is recovered in a clade with *Bohaiornis* in 16 of 22 (73%) studies, *Sulcavis* and *Zhouornis* in 18 of 23 (78%), *Shenqiornis* in 18 of 22 (82%), and *Parabohaiornis* in 19 of 21 (90%). Conversely, other taxa recovered in Bohaiornithidae tend to only rarely resolve near *Bohaiornis. Linyiornis* is recovered in a clade with *Bohaiornis* in 1 of 9 (11%) studies, *Eoenantiornis* in 3 of 21 (14%), and *Fortunguavis* in 4 of 19 (21%). *Gretcheniao* and *Musivavis*, both described as bohaiornithid-like but not bohaiornithid, are never recovered close to *Bohaiornis* (closest in *Liu et al., 2022*, where *Gretcheniao* is in a clade sister to the six original bohaiornithids plus BMNHC-Ph1204). *Beiguornis* has only been included in one study (*Wang, 2022a*), where it is recovered as a bohaiornithid, so it is tentatively included in the group pending future studies.

Phylogenetic tree topology

Extant bird phylogenetic trees in this study are trimmed versions of the maximum clade credibility supertree used in *Cooney et al., 2017*. This supertree follows *Jetz et al., 2012*, a tree time-scaled using Bayesian uncorrelated relaxed molecular clock data from 15 genes in 6663 avian species constrained by seven fossil taxa. These data were mapped onto the backbone of *Prum et al., 2015* which used Bayesian uncorrelated relaxed molecular clock data from 259 genes in 200 species constrained by 19 fossil taxa. The tree files and R code to merge them were taken from the supplement of *Cooney et al., 2017*.

All grafted bohaiornithid branch lengths were scaled linearly so that the total length of the avian portion of the tree was equal to 94 Ma following the estimate of *Kuhl et al., 2021*. The Ornithothoraces node was placed at 145 Ma after Bayesian morphological clock analysis of two independent character sets (*Cau, 2018*; *Wang et al., 2021a*). For the bohaiornithid portion of the tree, a topology

recovered by two recent analyses (*Wang et al., 2021b*; *Li et al., 2022*) was used where *Bohaiornis* and *Parabohaiornis* are sister taxa and the remaining bohaiornithids are in a basal polytomy. The position of *Beiguornis* is considered uncertain as the phylogeny in its description (*Wang, 2022a*) differs from this topology, but it was not included in any analyses where phylogeny was incorporated.

### Bohaiornithid fossil dating

*Shenqiornis* is referred to the Qiaotou Formation (*Wang et al., 2010*). This has been correlated to the Dawangzhangzi bed of the Yixian Formation (*Jin et al., 2008*), which is within the 'Upper Undivided Yixian Formation' of *Zhong et al., 2021* dated to 124 Ma. All other bohaiornithids are referred to the Jiufotang Formation. Most of these (*Wang, 2014a*; *Liu et al., 2022*) are from the Xiaotaizi/Lamadong locality dated to 119 Ma (*Yu et al., 2021*). *Longusunguis* IVPP V18693 from the Lingyuan locality (*Hu et al., 2020b*), *Zhouornis* BMNHC Ph756 from the Xiaoyugou locality (*Zhang et al., 2014*), and *Zhouornis* CNUVB 903 of uncertain provenance (*Zhang et al., 2013*) are referred to 121 Ma as the median age of the Jiufotang Formation (*Yu et al., 2021*).

## Taxonomic status of BMNHC-Ph1204

BMNHC-Ph1204 was previously identified to Bohaiornithidae indet. (*Liu et al., 2022*), though we herein refine the diagnosis to cf. *Sulcavis*. As noted in *Liu et al., 2022*, the large number of dentary teeth in BMNHC-Ph1204 is only seen in *Sulcavis* among known bohaiornithids, and the maxillary process of the nasal and interclavicular angle of the specimen are also consistent with those seen in *Sulcavis*. The final defining feature of BMNHC-Ph1204 noted by *Liu et al., 2022* is the lack of a fenestra in the maxilla, and the maxilla is poorly preserved in the *Sulcavis* holotype with no fenestra visible (*O'Connor et al., 2013*). Most of the synapomorphies of *Sulcavis* are present in BMNHC-Ph1204: teeth with a flat lingual margin creating a 'D-shaped' cross-section (*Liu et al., 2022* p.5); a broad nasal with a short, rostrally directed maxillary process (*Liu et al., 2022* p. 3); the caudal-most process of the synsacrum extending far caudally to the vertebra's articular surface (Figure 8 in *Liu et al., 2022*); a long and delicate acromion process on the scapula (Figure 7 in *Liu et al., 2022*); a convex lateral margin of the coracoid (Figure 8 in *Liu et al., 2022*); and an alular claw larger than that on the major digit (Figure 9 in *Liu et al., 2022*). The only missing synapomorphy is enamel with longitudinal ridges, autapomorphic for *Sulcavis* (*O'Connor et al., 2013*). BMNHC-Ph1204 is less mature than the *Sulcavis* holotype (*Figure 10*; *O'Connor et al., 2013*; *Liu et al., 2022*), and ontogenetic change in enamel ornamentation has been noted previously in reptiles (*Thies and Broschinski, 2001*; *Street et al., 2021*), so it is possible that this is an ontogenetic difference. The same can be said of the additional carpal element in BMNHC-Ph1204, as carpometacarpal fusion is believed to occur relatively late in bohaiornithid ontogeny (*Wang, 2014a*; *Liu et al., 2022*). Therefore, with this uncertainty from ontogeny, we refer BMNHC-Ph1204 to cf. *Sulcavis*. Should this specimen later be referred to its own taxon, the similarities above imply the new taxon would be very closely related to *Sulcavis*, and this specimen would still be the most appropriate for filling in missing pieces of the skull of *Sulcavis*, as in *Figure 2*.

## Bohaiornithid skull reconstruction for MA and FEA

Reconstructions of bohaiornithid skulls are provided in *Figure 2*. No complete bohaiornithid skulls are currently known, so digital reconstruction was necessary to create workable biological models (*Lautenschlager, 2016*). Reconstructions of a given taxon used material from that taxon where possible, but each reconstruction required at least one bone from another taxon. Ideally these replacement bones would come from a taxon's closest relative, but the relationships within Bohaiornithidae are unstable aside from the sister relationship of *Bohaiornis* and *Parabohaiornis*. For reconstruction purposes, we treated *Shenqiornis* and *Sulcavis* as sister taxa due to them having the next most frequent sister pairing among bohaiornithids (*Wang et al., 2022b*; *Zhang et al., 2014*; *Wang et al., 2014b*) and the superficially similar shallow angle of their rostra. *Longusunguis* and *Zhouornis* have the most unstable placement within Bohaiornithidae, so supplemental parts were taken from taxa arbitrarily. Fortunately these taxa each had two moderately complete skulls, so little material from other taxa was needed.

Only *Bohaiornis* IVPP V17963 preserves the posteroventral region of the skull, so this region in all reconstructions is based on this. This affects jaw-opening mechanical advantage (OMA) of the upper jaw, and thus this functional index should be interpreted cautiously. We interpret the bone in *Bohaiornis* LPM B00167 previously labelled as the postorbital (*Hu et al., 2011*) to be the quadratojugal, as its

shape is more similar to the short and weakly forked quadratojugal typical of enantiornithines (*Wang and Hu, 2017a*) than the elongate postorbital of *Longusunguis* IVPP V18693 (*Hu et al., 2020b*), *Parabohaiornis* IVPP V28398 (*Wang, 2023*), *Sulcavis* BMNHC Ph-805 (*O'Connor et al., 2013*), and cf. *Sulcavis* BMNHC Ph-1204 (*Liu et al., 2022*). The alternative is that this bone is in fact the postorbital, meaning *Bohaiornis* (and possibly *Shenqiornis* and *Zhouornis*) lacked a complete postorbital bar.

The jugal and postorbital bones of *Longusunguis* are elongate and believed to be in contact in life (*Hu et al., 2020b*), which would increase the overall rigidity and stability of the skull. This is also very likely in *Parabohaiornis* and *Sulcavis*, which preserve both a dorsally curving jugal and elongate postorbital (*Figure 2C and D*). We reconstruct these bones as also in contact in other bohaiornithids (*Figure 2A, E and F*), though this is less certain. Notably if we are incorrect in our reinterpretation of a bone in LPM B00167 identified as the postorbital (*Hu et al., 2011*) as the quadratojugal, then this much smaller element could not contact the jugal in life.

## Bohaiornithid ontogeny

Ontogeny is an important dimension to account for when discussing the morphology and ecology of extinct animals, but the ontogeny of Bohaiornithidae (and enantiornithines in general) is poorly understood. *Hu and O'Connor, 2017* devised a useful 'character stage' system for tracking enantiornithine maturity based on the fusion of compound bones. Bohaiornithids are aged using this framework in *Figure 10*. *Hu and O'Connor, 2017* combine the fusion of the proximal metatarsus, the tibiotarsus, and carpometacarpus into a single stage 3 (presumably due to a lack of resolution). *Longusunguis* IVPP V18693 has a fused tibiotarsus and unfused tarsometatarsus and carpometacarpus and *Sulcavis* BMNHC-Ph805 has a fused tibiotarsus and tarsometatarsus but unfused carpometacarpus, indicating the tibiotarsus fused first and the carpometacarpus last within this family. We thus subdivide the stage 3 of *Hu and O'Connor, 2017* into 3a (formation of the tibiotarsus), 3b (proximal fusion of the tarsometatarsus), and 3c (fusion of the carpometacarpus) in bohaiornithids.

The above stages only approximate biological maturity, however. Histological data from two bones has been used to suggest that *Zhouornis* CNUVB-0903 had achieved sexual maturity but not skeletal maturity (*Zhang et al., 2013*) A recent study (*Atterholt et al., 2021*) has shown that histological maturity can vary greatly within individual enantiornithines but upholds the diagnosis of CNUVB-0903 as an adult. The indeterminate bohaiornithid CUGB P1202 is considered the least mature published

**Table 10.** Comparison of the bone character stages of *Hu and O'Connor, 2017* with the histological maturity stages of *Atterholt et al., 2021*.
Subadult maturity appears to be obtained by stage 1 (the most immature fusion stage of any specimen examined in this work), while full skeletal maturity is not obtained until after stage 3 (the most mature fusion stage). Histological maturities are identified by *Atterholt et al., 2021* directly, bone fusion stages are taken from *Hu and O'Connor, 2017* where possible and intuited based on their fusion criteria where not.

| Specimen | Atterholt et al. Histological maturity | Hu and O'Connor Character stage |
|---|---|---|
| *Parapengornis* IVPP V18687 | Young subadult | 1 |
| *Eopengornis* STM24-1 | Young subadult | 2a |
| *Monoenantiornis* IVPP V20289 | Subadult | 2a |
| *Parvavis* IVPP V18586 | Subadult | 2b |
| *Cruralispennia* IVPP V21711 | Subadult | 3 |
| *Pterygornis* IVPP V16363 | Young adult | 3 |
| *Avimaia* IVPP V25371 | Adult | 3 |
| *Mirarce* UCMP139500 | Adult | 3 |
| *Mirusavis* IVPP V18692 | Adult | 3 |
| STM 29-8 | Adult | 3 |
| *Zhouornis* CNUVB-0908 | Adult | 3 |

member of the family (*Peteya et al., 2017*). It was proposed as sexually mature due to its elongate tail feathers (with the assumption they served a sexual display purpose; *Peteya et al., 2017*), but similar feathers in decidedly juvenile enantiornithines (STM 34-7 [*Zheng et al., 2012*] and 34-9, IVPP V15564 [*O'Connor et al., 2020a*]) call this into doubt. By correlating the histological maturity of *Atterholt et al., 2021* with the character stage system (*Table 10*), we see that subadulthood begins at or before character stage 1 and full maturity occurs after completion of character stage 3. We thus consider all bohaiornithids aside from the two mentioned above to be mature subadults or young adults, as proposed previously (*Wang, 2014a*; *Liu et al., 2022*; *Hu et al., 2020b*). Most extant animals are considered 'mature' upon reaching the subadult stage (*Hone et al., 2016*), so all specimens examined herein should be adequately mature for comparison to mature extant birds.

## Extant sampling

Extant bird masses were taken from the bird ecology database AVONET (*Tobias et al., 2022*) and combined with bird diet data from EltonTraits 1.0 *Wilman et al., 2014* following the diet cut-offs in *Table 2*. The mass dataset consists of 8758 birds: 169 folivores, 931 frugivores, 1122 generalists, 475 granivores, 5061 invertivores, 450 nectarivores, 207 piscivores, 35 scavengers, and 308 tetrapod hunters.

No additional extant birds are sampled for MA or FEA beyond the dataset of *Miller et al., 2023*, meaning these datasets consist of 141 birds: 9 folivores, 17 frugivores, 17 generalists, 8 husking granivores, 8 swallowing granivores, 43 invertivores, 7 nectarivores, 15 piscivores, 8 scavengers, and 9 tetrapod hunters.

Five additional taxa were added to the ungual traditional morphometric (TM) dataset of *Miller et al., 2023* to increase the sample of anisodactyl non-raptorial perching birds. Target taxa were identified from the literature as birds with well-recorded perching behaviour and no record of grasping or manipulating prey with the pes. This list of target taxa was sent in a formal loan request to Serina Brady of Carnegie Museum of Natural History, who provided photographs of all available specimens fitting the criteria. In total, the extant TM dataset includes 66 birds: 9 ground birds, 23 non-raptorial perching birds, 4 shrikes, 18 raptors taking large prey, and 12 raptors taking small prey.

Of the 408 extant bird samples included in the skull TM analysis of *Clark et al., 2023*, 61 were removed due to not meeting the definition of any diet category in *Table 2* (they were 'in between' diets as defined here, often with 50% of the diet from two sources). Thus, the final skull morphometric dataset consisted of 347 birds across 298 species: 9 folivores, 40 frugivores, 53 generalists, 5 granivores, 175 invertivores, 26 nectarivores, 27 piscivores, 4 scavengers, and 8 tetrapod hunters.

## Analytical techniques

### Body mass

The granivore diet category was not split into husking and swallowing granivores; past work (*Miller et al., 2022*; *Miller et al., 2023*) did not find the two subcategories to meaningfully differ in body mass. Average masses of groups were compared via phylogenetic HSD (pairwise function of R package RRPP [*Collyer and Adams, 2018*] used to compare group means). Diagnostic cut-off values between groups were found with the R package OptimalCutpoints (*López-Ratón, 2014*) version 1.1-5 by optimising the Youden index, with bootstrap estimation of 95% confidence intervals using the R package boot (*Davison and Hinkley, 1997*) version 1.3-28.

### Pedal traditional morphometrics, mechanical advantage, and finite element analysis

The extant MA and FEA data in this study is unchanged from *Miller et al., 2023*, and new data collected from fossil specimens followed the same procedures as *Miller et al., 2023*. New fossil MA data were measured from the reconstructions in *Figure 2* in CorelDraw X8. FEA models based on the reconstructions in *Figure 2* were created and solved within HyperWorks 2022 Student Edition (*HyperMesh* and *Optistruct*, Altair Engineering, Inc, USA). We make the assumption that toothed and beaked models are directly comparable given that both operate under constraints of preventing breakage and deformation to the jaw (*Miller and Pittman, 2021*). Presence of a rhamphotheca is known to reduce stress experienced during a bite (*Lautenschlager et al., 2013*), so the beaked models all include a modelled rhamphotheca (*Miller et al., 2022*) to account for this effect. Pedal

TM data are measured from scale photographs where available, though the five new included taxa were photographed without scale. All digits from a single individual were measured from a single photograph, so their relative scale is preserved. Our TM data is based solely on angles and length ratios, so this lack of scale should not be an issue. Pedal ecological categories for TM follow *Miller et al., 2023*; of note, we define raptors taking large prey as those with records of regularly taking prey which cannot be completely encircled by the pes, and raptors taking small prey as those without such records. As pointed out in *Fowler et al., 2009*, this division commonly follows phylogenetic lines, but we observe exceptions such as *B. virginianus* taking large prey (*Artuso, 2020*) and *Buteogallus anthracinus* not taking large prey (*Schnell, 2020*) clustering with birds in the same pedal ecological group rather than their phylogenetic relatives. All analyses of the data (PCA, FDA, pFDA, ancestral state reconstruction, etc.) were performed in R version 4.1.2 (*R Development Core Team, 2020*), with scripts and raw data including measurements and FE models available from Mendeley Data (https://doi.org/10.17632/7xtpbv27zh.3).

## Skull traditional morphometrics

*Clark et al., 2023* recently used skull traditional morphometrics to great effect in both differentiating extant bird diets and diagnosing the diet of longipterygid enantiornithines. Thus, we incorporate their data and methodology here as well to expand the evidence supporting our diet diagnoses. Measurements of skull and rostrum length for extant birds are unchanged from *Clark et al., 2023*. *Clark et al., 2023* do not diagram their measurement landmarks, and the caudal landmark of the rostral length as 'the caudal margin of the lacrimal (i.e. the rostral margin of the orbit)' does not specify a dorsoventral height. So, for all new rostrum length measurements of bohaiornithids, we measured to the dorsoventral midpoint of the caudal margin of the ventral ramus of the lacrimal.

Diet assignments of extant taxa were changed from those used by *Clark et al., 2023* to those in *Table 2* for consistency, following information in EltonTraits 1.0 (*Wilman et al., 2014*). We also use mass estimates for taxa from AVONET (*Tobias et al., 2022*), rather than those listed by *Clark et al., 2023*, for consistency within this work.

*Clark et al., 2023* visualise (their Figure 4) and generally interpret their skull morphometrics in terms of proportional rostral length (i.e. rostrum length divided by skull length) vs. $log_{10}$ skull length. We have modified both of these factors in *Figure 8*. Firstly, we $log_{10}$-transform both rostrum and skull length before taking their ratio, as this better normalises the distribution of proportional rostral length. Actual distribution of the data along this axis is minimally affected. Secondly, we use a relative skull length instead of absolute skull length. Absolute skull length is essentially a size proxy; *Figure 8—figure supplement 1* shows skull length and body mass creating similar distributions, with the position of diet groups maintained relative to one another (while spreading more evenly along the x-axis when using body mass). This work already discusses body size as a diet proxy at length, so including it in morphometric discussion would be redundant. Through $log_{10}$-$log_{10}$ regression, we found that skull length generally increases with body mass with a scaling exponent of 2.75 (slightly negative allometry, $R^2 = 0.63$) for the extant birds sampled. Thus we calculated relative skull length as:

$$relative\ skull\ length = \frac{\left(log_{10}\left(skull\ length\right)\right)^{2.7482}}{log_{10}\left(body\ mass\right)}$$

Relative skull length approximates how much longer or shorter a given bird's skull is than the expected length for a bird of the same body mass.

## Ancestral state reconstruction

The backbone for the enantiornithine tree topology used in ancestral state reconstruction follows *Wang et al., 2021b*. This tree contains 34 of the 56 species used in ancestral state reconstruction. Additional taxa were grafted onto this backbone following *Wang et al., 2022b*; *Liu et al., 2022*; *Wang, 2022a*; *O'Connor, 2009*; *Li et al., 2022*; *O'Connor et al., 2020b*; *Li et al., 2012a*; *Kurochkin et al., 2013*; and *Atterholt et al., 2018*. Some taxa have only been included in one phylogenetic analysis, which placed said taxa within Bohaiornithidae, Longipterygidae, or Pengornithidae (e.g. *Noguerornis* placed within Longipterygidae by *O'Connor, 2009*). These taxa all lack the synapomorphies used to formally define the relevant clade, and their placements within them are likely artefacts of

sharing a few characters while most data are missing. We therefore maintained monophyly of Bohaiornithidae, Longipterygidae, and Pengornithidae by placing these ambiguous taxa as sister to the family they were placed within. Time-scaling was performed using the R package paleotree (*Bapst, 2012*). All enantiornithine taxa were placed at their earliest occurrence, with species divergence arbitrarily chosen to take 1000 years. The enantiornithine phylogeny was rooted at 144 Ma after *Cau, 2018* and *Wang et al., 2021a*. Ages of localities follow *Zhong et al., 2021*; *Yu et al., 2021*; *Chen et al., 2006*; *Fregenal-Martínez, 2017*; *Lockley et al., 2018*; *Porfiri et al., 2018*; *Chen et al., 2020*; *Yang et al., 2020*; *Napoli et al., 2021*; *Galobart et al., 2022*; *Montano et al., 2022*; and *Ramezani et al., 2022*, with precise notes in the data repository (https://doi.org/10.17632/7xtpbv27zh.3). If the horizon of a specimen was in doubt, it was placed at the median age of the formation.

The diet of the common ancestor of Enantiornithes was reconstructed in two ways: qualitatively, directly labelling diets onto taxa to reconstruct the diet of ancestral nodes; and quantitatively, reconstructing ancestral body mass, jaw MA and functional indices, jaw FEA MWAM strain and interval values, and pedal TM values, and predicting ancestral diet from these reconstructed values. The qualitative set includes quantitative diet reconstructions made here, quantitative reconstructions from *Miller et al., 2022* and *Miller et al., 2023*, all qualitative hypotheses of diet in enantiornithines (*Table 11*), and all enantiornithines with mass data (approximating enantiornithines known from substantial fossil material, as they need to be somewhat complete for mass estimation; *Cuspirostrisornis*, *Dalingheornis*, *Gracilornis*, *Jibeinia*, *Longchengornis*, *Microenantiornis*, and *Paraprotopteryx* were excluded due to never being included in a phylogenetic analysis). The latter were coded with 'Unknown' diet to illustrate uncertainty in the reconstruction. As hypothesised diets are less precise than the diet categories used elsewhere in this study, they were lumped together as herbivores (folivores, frugivores, granivores, and nectarivores), invertivores, omnivores (generalists), or vertivores (piscivores, tetrapod hunters, and scavengers). In total, 14 of 30 (47%) of diets in this reconstruction are supported by preserved meal evidence or quantitative diet proxies. A total of 56 taxa are included in the qualitative ancestral state reconstruction.

Quantitative ancestral state reconstruction includes data from the present study and our two other quantitative analyses of enantiornithines (*Miller et al., 2022*; *Miller et al., 2023*), as well as pedal TM data for *Fortunguavis* from *Pittman et al., 2022* and mass data from *Miller and Pittman, 2021* and *Serrano et al., 2015*. Only MWAM strain of FEA models was reconstructed, not full intervals data, as we found the validity of reconstructing an individual strain interval to be dubious. If pedal TM data was available for multiple specimens of a genus, we favoured the most mature specimen to represent the genus. If mass data was available for multiple specimens of a genus, we favoured the largest mass estimate as an assumption of it representing the most mature sample. The large-toothed *Longipteryx* morphotype (*Miller et al., 2022*) was assumed more mature than the small-toothed morphotype and used for all reconstructions. A total of 44 taxa are used in ancestral state reconstruction of body mass, 13 in upper jaw MA, 9 in lower jaw MA, 13 in FEA, and 16 in pedal TM.

*Sapeornis* was used as the outgroup in ancestral state reconstruction. Mass data for the taxon was taken from *Serrano et al., 2015*. MA data was measured from the reconstruction in *Hu et al., 2020a*. FEA data was taken from a model constructed by Yuen Ting (Athena) Tse for an in-progress collaborative work based on the same reconstruction (*Hu et al., 2020a*). TM data were taken from *Pittman et al., 2022*. Its qualitative diet was classified as herbivorous based on preserved seed meals (*O'Connor, 2019a*). Its divergence from Enantiornithes was placed at 147 Ma after *Wang et al., 2021a*.

Ancestral states were estimated using the fastAnc() function in Phytools 0.7-90 (*Revell, 2012*). As polytomies are present in the enantiornithine tree used, ancestral states were calculated 10,000 times with random resolutions of the polytomies and averaged. As some diets were uncertain (e.g. *Zhouornis* may be either herbivorous or a generalist), and fastAnc() cannot accept probabilistic qualitative states for terminal taxa, we coded diets quantitatively as percent likelihood for each diet.

## Acknowledgements

We thank Serina Brady, Chase Mendenhall, Stephen Rogers (Carnegie Museum of Natural History), Andrew Kratter, and David Steadman (Florida Museum of Natural History) for their assistance and expertise in selecting physical specimens for this study. We would also like to thank Kathryn C Gamble DVM, MS, Dipl ACZM, Dip ECZM (ZHM), Veterinary Advisor Coraciiformes/Bucerotiformes TAG for

**Table 11.** Qualitative dietary hypotheses for enantiornithine taxa.

The hypothesised diet listed may overgeneralise the hypothesis of the original publication for the sake of brevity. Most notably we often simplify hypotheses like 'carnivory taking small animals' to invertivory, as when these authors provide analogous extant birds for context, they predominantly were invertivorous birds.

| Taxon | Hypothesised diet | Reference |
|---|---|---|
| *Avisaurus archibaldi* | Raptorial | *Martyniuk, 2012* |
| *Bohaiornis guoi* | Durophagy, invertivory, raptorial | *Chiappe and Meng, 2016*; *Zhou et al., 2021*; *Benito and Olivé, 2022*; *Li et al., 2012b* |
| *Boluochia zhengi* | Piscivory, raptorial | *Benito and Olivé, 2022*; *Yang et al., 2020*; *Zhou, 1995*; *Hou, 1997*; *Feduccia, 1999*; *Chiappe and Walker, 2002* |
| *Brevirostruavis macrohyoideus* | Invertivory, nectarivory | *Benito and Olivé, 2022*; *Li et al., 2022* |
| *Cathayornis yandica* | Invertivory | *Martyniuk, 2012* |
| *Concornis lacustris* | Invertivory | *Martyniuk, 2012* |
| *Cuspirostrisornis houi* | Raptorial | *Martyniuk, 2012* |
| *Chiappeavis magnapremaxillo* | Invertivory | *Benito and Olivé, 2022*; *Lockley et al., 2018* |
| *Dapingfangornis sentisorhinus* | Piscivory | *Martyniuk, 2012* |
| *Falcatakely forsterae* | Frugivory | *Benito and Olivé, 2022* |
| *Gettyia gloriae* | Scavenging | *Benito and Olivé, 2022* |
| *Gobipipus reshetovi* | Granivory, invertivory | *Chatterjee, 2015* |
| *Gobipteryx minuta* | Granivory, invertivory | *Chatterjee, 2015*; *Benito and Olivé, 2022* |
| *Halimornis thompsoni* | Piscivory | *Martyniuk, 2012*; *Benito and Olivé, 2022* |
| *Longipteryx chaoyangensis* | Piscivory | *O'Connor and Chiappe, 2011a*; *Chiappe and Meng, 2016*; *Zhang et al., 2001*; *Wang et al., 2015b*; *O'Connor, 2009*; *Martyniuk, 2012*; *Chatterjee, 2015*; *Benito and Olivé, 2022* |
| *Longirostravis hani* | Invertivory, probing | *Chiappe and Meng, 2016*; *Martyniuk, 2012*; *Chatterjee, 2015*; *Benito and Olivé, 2022*; *Hou et al., 2004* |
| *Mirarce eatoni* | Raptorial | *Benito and Olivé, 2022* |
| *Neuquenornis volans* | Raptorial | *Martyniuk, 2012*; *Chatterjee, 2015*; *Benito and Olivé, 2022*; *Chiappe and Walker, 2002* |
| *Parabohaiornis martini* | Invertivory | *Chiappe and Meng, 2016* |
| *Pengornis houi* | Hard invertivory, soft invertivory | *O'Connor and Chiappe, 2011a*; *Chiappe and Meng, 2016*; *Martyniuk, 2012*; *Chatterjee, 2015*; *Benito and Olivé, 2022* |
| *Rapaxavis pani* | Invertivory, probing | *Chiappe and Meng, 2016*; *Martyniuk, 2012*; *Chatterjee, 2015*; *Benito and Olivé, 2022* |
| *Shanweiniao cooperorum* | Invertivory, probing | *Chiappe and Meng, 2016*; *Benito and Olivé, 2022* |
| *Shengjingornis yangi* | Invertivory, probing | *Benito and Olivé, 2022* |
| *Shenqiornis mengi* | Durophagy, invertivory | *O'Connor and Chiappe, 2011a*; *Martyniuk, 2012*; *Benito and Olivé, 2022*; *Wang et al., 2010* |
| *Sinornis santensis* | Carnivory, folivory | *Martyniuk, 2012*; *Feduccia, 1999*; *Chiappe and Walker, 2002* |
| *Soroavisaurus australis* | Raptorial | *Martyniuk, 2012*; *Chatterjee, 2015*; *Chiappe and Walker, 2002* |
| *Sulcavis geeorum* | Durophagy | *O'Connor et al., 2013*; *Chiappe and Meng, 2016*; *Benito and Olivé, 2022* |
| *Vescornis hebeiensis* | Invertivory | *Martyniuk, 2012* |
| *Zhouornis hani* | Invertivory | *Chiappe and Meng, 2016* |

providing coraciiform radiographs used in this study. We also thank Gavin Thomas and Ryan Felice for their insight in constructing the consensus phylogeny of extant birds. Finally, we thank Yuen Ting (Athena) Tse for constructing the FEA model of *Sapeornis* used as an outgroup in ancestral state reconstruction. CVM is supported by a Postgraduate Scholarship from The University of Hong Kong (HKU PGS). MP is supported by the Research Grant Council of Hong Kong's General Research Fund (17120920; 17103315; 17105221) and the School of Life Sciences at The Chinese University of Hong Kong. XW is supported by the Taishan Scholars Program of Shandong Province (Ts20190954).

## Additional information

### Funding

| Funder | Grant reference number | Author |
| --- | --- | --- |
| Postgraduate Scholarship, The University of Hong Kong | | Case Vincent Miller |
| Research Grant Council of Hong Kong's General Research Fund | 17120920 | Michael Pittman |
| School of Life Sciences, The Chinese University of Hong Kong | | Michael Pittman |
| Taishan Scholars Program of Shandong Province | Ts20190954 | Xiaoli Wang |
| Research Grant Council of Hong Kong's General Research Fund | 17103315 | Michael Pittman |
| Research Grant Council of Hong Kong's General Research Fund | 17105221 | Michael Pittman |

The funders had no role in study design, data collection and interpretation, or the decision to submit the work for publication.

### Author contributions

Case Vincent Miller, Conceptualization, Software, Formal analysis, Investigation, Visualization, Methodology, Writing - original draft, Writing - review and editing; Jen A Bright, Conceptualization, Formal analysis, Supervision, Investigation, Methodology, Writing - review and editing; Xiaoli Wang, Xiaoting Zheng, Resources; Michael Pittman, Conceptualization, Resources, Formal analysis, Supervision, Funding acquisition, Investigation, Visualization, Methodology, Writing - review and editing

### Author ORCIDs

Case Vincent Miller http://orcid.org/0000-0002-6467-0199
Jen A Bright http://orcid.org/0000-0002-9284-9591
Michael Pittman http://orcid.org/0000-0002-6149-3078

Reviewer #1 (Public Review): https://doi.org/10.7554/eLife.89871.3.sa1
Reviewer #2 (Public Review): https://doi.org/10.7554/eLife.89871.3.sa2
Reviewer #3 (Public Review): https://doi.org/10.7554/eLife.89871.3.sa3
Author response https://doi.org/10.7554/eLife.89871.3.sa4

## Additional files

### Supplementary files
• MDAR checklist

- Supplementary file 1. K and Kmult values for each extant dataset.
- Supplementary file 2. K values for individual MA and functional index variables of the extant skull dataset.
- Supplementary file 3. Significant differences between extant diet groups, based on phylogenetic HSD of MA and functional indices.
- Supplementary file 4. Sensitivity analysis of quadrate position affecting bohaiornithid predictions diet by FDA from MA and functional indices of extant bird jaws.
- Supplementary file 5. Significant differences between extant diet groups, based on phylogenetic HSD of FEA intervals data.
- Supplementary file 6. K values for individual variables used in pedal TM analyses.
- Supplementary file 7. Significant differences between extant pedal ecology groups, based on phylogenetic HSD of pedal TM data.
- Supplementary file 8. K values for individual variables used in skull TM analyses.

## Data availability

The raw data collected in this study, the outputs of finite element analysis, and all code used in data analysis and figure creation are deposited at the https://doi.org/10.17632/7xtpbv27zh.3.

The following dataset was generated:

| Author(s) | Year | Dataset title | Dataset URL | Database and Identifier |
|---|---|---|---|---|
| Miller CV, Bright JA, Wang X, Zheng X, Pittman M | 2024 | Synthetic analysis of trophic diversity and evolution in Enantiornithes with new insights from Bohaiornithidae | https://doi.org/10.17632/7xtpbv27zh.3 | Mendeley Data, 10.17632/7xtpbv27zh.3 |

The following previously published dataset was used:

| Author(s) | Year | Dataset title | Dataset URL | Database and Identifier |
|---|---|---|---|---|
| Tobias JA, Sheard C, Pigot AL, Devenish AJM, Yang J, Sayol F | 2023 | AVONET: morphological, ecological and geographical data for all birds | https://figshare.com/s/b990722d72a26b5bfead | figshare, b990722d72a26b5bfead |

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
