## [Editor Report · eLife assessment]

This **important** study explores numerous lines of evidence for the surprisingly diverse diets of a group of toothed birds that lived over 100 million years ago. The large amount of data the authors collected forms a **solid** dataset. The methods might in principle be extensible to other limbed vertebrates, although there are concerns regarding some of the details. The article will be of interest to colleagues studying ecological evolution in birds or dinosaurs more generally, as well as to anyone studying the impact of the mass extinction event 66 million years ago.

---

## [Referee Report · Reviewer #1 (Public Review)]

Understanding the ecology including the dietary ecology of enantiornithines is challenging by all means. This work explores the possible trophic diversity of the "opposite-bird" enantiornithines by referring to the body mass, jaw mechanical advantage, finite element analysis of the jaw bones, and morphometrics of the claws and skull of both fossil and extant avian species. By incorporation the dietary information of longipterygids and pengornithinds, the authors predicted a wide variety of foods for enantiornithine ancestors. This indicates the evolutionary successes of enantiornitine during Cretaceous is very likely to have been driven by the wide range of recipes. I believe this work represented the most comprehensive analysis of enantiornithines' diet and trophic diversity by far and the first systematic dietary analysis of bohaiornithids, though the analysis themselves are largely based on the indirect evidence including jaw bone morphologies and claw and skull morphometrics. Anyway, I believe the authors did most the paleontologists could do, and I do not know whether the conclusions could be further supported by incorporating some geochemical data, as most of the specimens the authors analyzed were recovered from a small geographic area. The results also indicate that the developmental trajectories of enantiornithines, at least for jaw bones, might also have been diverse to some extent in response to the diverse ecological niches they adapted. My only concern regarding the analysis is to what extent the conclusions are convincing by comparing specimens representing various ontogenetic stages. This concern has been addressed in the revised manuscript. I believe the authors have almost exhausted all available methods, and I congratulate the authors for the detailed study they conducted.

---

## [Referee Report · Reviewer #2 (Public Review)]

Miller et al. take a variety of measurements and analytical techniques to assess the ecology of various species of the enantiornithine clade Bohaiornithidae. From this they suggest that the ancestral enantiornithine was a generalist and that the descendant clades occupied a breadth of niches similar to that of the radiation of derived birds after the K-Pg extinction.

Overall, I find the idea that enantiornithines had occupied a similar niche breadth to post-K-Pg derived birds to be a curious, thought-provoking proposal.

I am satisfied with the edits made by the authors and approve the revised version of the manuscript.

---

## [Referee Report · Reviewer #3 (Public Review)]

Summary:

The authors use several quantitative approaches to characterize the feeding ecologies of bohaiornithid enantiornithines, including allometric data, mechanical advantage and finite element analyses of the jaw, and morphometric analyses of the claws. The authors combine their results with data for other enantiornithines collected from the literature to shed new insight on the ecological evolution of Enantiornithes as a clade.

Although the authors have taken steps to improve their paper, I generally find improvements unsatisfying, especially regarding my comments.

My remaining concerns:

Teeth: My concern here is not whether having teeth limits available niche space compared to having a keratinous beak. Rather, my concern regards how exploitation of the same niche space might be differently reflected in parameter space between birds with teeth and birds with beaks. Can we reliably expect two species that both eat seeds to occupy the same parameter space if, for example, distribution of stress/strain is across a series of teeth vs. across a more uniform beak? In this manuscript, the authors are clearly making this assumption, but that assumption is not made explicit, let alone justified. The authors should discuss this.

Cranial kinesis: As with teeth, my concern here regards our ability to compare data between birds with and without a flexible beak to mitigate forces when foraging. I appreciate that the functional complexity of the kinetic neognath skull precludes our ability to account for it in analyses such as these, but when comparisons are made using these analyses *specifically among neognaths*, we can reliably assume that we are comparing like to like - that is, we can assume that both have kinetic skulls, and so kinesis is reflected similarly in the data for each bird. Similarly, even if a comparison between two neognaths focuses exclusively on the mandible - in which cranial kinesis is not directly reflected - we can assume that those mandibles serve as comparisons between functionally similar systems. However, we cannot necessarily make those same assumptions when comparing the kinetic skull of a neognath to the akinetic skull of an enantiornithine. Indeed, even when focusing just on the mandible, can we reliably assume that data collected from an akinetic enantiornithine reflect the same comparative context as data collected from kinetic neognaths? I appreciate that the authors added a call for better functional understanding of bird cranial kinesis - a call I enthusiastically endorse - but the authors should still discuss how that current lack of understanding impacts interpretations of the comparisons they draw.

Finally, I still find the discussion to be overly long and lacking clear focus and organization. I again urge the authors to minimally consider adding subheadings to better allow the reader to follow the flow of ideas, and I second Reviewer #1's suggestion to add a "Conclusions" section.

---

## [Author Response]

The following is the authors’ response to the current reviews.

We thank the reviewers for their valuable feedback which has improved this work greatly from its original form, and are elated to have such glowing reviews of the revised work published alongside the revised preprint. Reviewer 3 raises some final salient points, which deserve a brief address here.

Teeth: We thank the reviewer for clarifying their points. We do make the assumption that the ecological parameter space of toothed and beaked organisms will be comparable. Both are governed by the same set of physical principles and have the jaw bone as the most likely point of failure (teeth are harder than bone, and keratinous rhamphothecae are malleable and can be regrown with relative ease when deformed). Differences in stress/strain distribution between toothed and beaked organisms will occur but are already accounted for in our methods as we model both the teeth and rhamphotheca and will observe these different effects. We have added an explicit statement of this hypothesis to the Methods section of the manuscript.

Cranial kinesis: In our opinion, it is a safe assumption that the lower jaws of extant birds and enantiornithines are comparable. We do not see why the acquisition of kinesis in the upper jaw would generally affect the functional role of or constraints on the lower jaw. One possibility we discussed is that a quickly-moving kinetic premaxilla could let the lower jaw move a shorter distance during effective prey capture and lower the selection for speed (i.e. allow jaw-closing MA to remain higher). While we have added this possibility to our call for the investigation of cranial kinesis, we consider it too speculative to begin altering interpretations of fossil taxa. All raw measurement data remains available so that, if evidence is found for cranial kinesis having predictable effects on our measured parameters, future researchers can re-analyse our data and update any ecological predictions accordingly.

Organization: To our knowledge eLife format incorporates what one would think of as a Conclusions section into the Discussion. Our Discussion section currently contains 18 subheadings which should guide a reader to any specific topic of interest. The Discussion also progresses from a more narrow to broad focus which we and several colleagues find intuitive.

We thank all three reviewers once again for their feedback that has improved this work and their kind words throughout the process.

The following is the authors’ response to the original reviews.

We thank all three reviewers for their detailed reviews, and generally agree with their feedback. To accompany the reviewed preprint of this manuscript, we wished to respond to comments from the reviewers so that they (and the public) will know what we are planning to incorporate in the revised manuscript we are currently preparing. If there are any comments on our plans in the meantime, please let us know.

• Reviewer 1, on concerns regarding identification of ontogenetic stage and comparison of taxa from different ontogenetic stages: It is fair to say that enantiornithine ontogeny is still poorly understood, though we believe all current evidence points to each specimen used in this study to being adequately mature for comparison to the extant birds used in the study. Stages of skeletal fusion are the standard method of assessing enantiornithine ontogeny (Hu and O'Connor 2017), and our comparison of histological work (Atterholt, Poust et al. 2021) to skeletal stages in Table S4 suggests a transition from juvenile to subadult in stage 0 or 1 and from subadult to adult within stage 3. Thus, the specimens we quantitatively examine in this study, all at stages 2 or 3 (Figure S10), are advanced subadults or adults. It is well-known that many living animals considered “adults” would be considered subadults or even juveniles to a palaeontologist (Hone, Farke et al. 2016). So, even if some individuals in this study are not fully skeletally mature, they should have obtained the morphology which they would possess for most of their lives and thus the morphology which undergoes selective pressure. We will add this context to the “Bohaiornithid Ontogeny” section and thank the reviewer for seeking more detail for this point.

• Reviewer 2, on need of a context figure: We have an artistic life reconstruction of a bohaiornithid in preparation, and can include that in the revised manuscript as a figure.

• Reviewer 2, on raptor claw categories: We explain these categories in-depth in a previous work (Miller, Pittman et al. 2023). However, we will now add a short summary of that explanation to this work so that this manuscript will become self-contained in this regard. In short, the “large raptor” category includes extant birds with records of regularly taking prey which cannot be encircled with the pes, while birds in the “small raptor” have no such records. As Reviewer 2 points out this does often follow phylogenetic lines, but not always. E.g. most owls specialise in taking small prey, but the great horned owl Bubo virginianus regularly takes mammals and birds larger than its pes (Artuso, Houston et al. 2020); and conversely we can only find reports of the common black hawk Buteogallus anthracinus taking prey samll enough for the pes to encircle (Schnell 2020) despite other accipiters frequently taking large prey. In both cases these taxa plot in PCA nearer to other large or small raptors (respectively) than to their phylogenetic relatives.

• Reviewer 3, on teeth vs beaks: We are not aware of any foods which are exclusive to toothed or beaked animals. There are some aspects of extant bird biology that may affect the way a certain diet may need to be adapted to which we do comment on, e.g. discussion of alternatives to the crop and ventriculus for processing plant matter in the Bohaiornithid Ecology and Evolution section. For functional studies, e.g. FEA, we have included the rhamphotheca in toothless models which serves the same role as teeth, to be a feeding surface. It should not matter, in theory, if the feeding surface is hard or soft as mechanical failure occurs in high stress/strain states regardless of the medium. If having teeth necessarily increases or decreses overall stress/strain relative to a beak (and from our work this does not appear to be the case), this would in turn necessarily limit dietary options. So, all models in our work should be directly comparable.

As an additional note on this topic, we address tooth shape in bohaiornithids at the end of the Bohaiornithid Ecology and Evolution section. We specifically note that their tooth shape is likley controlled by phylogeny in the current version, though we will add a note in the upcoming version that the morphospace of bohaiorntihid teeth overlaps that of many other clades with purportedly diverse diets, which is consistent with a hypothesis of diverse diets within the clade.

• Reviewer 3, on cranial kinesis: Our FE models should be unaffected by cranial kinesis, as these are two-dimensional and model the akinetic lower jaw only. Some mediolateral kinesis may be relevant in the mandible in the form of “wishboning” in different taxa, but its prevalence in extant birds is currently unknown. The preservation of enantiornithines (two-dimensionally and typically in lateral view) limits the ability to capture any mediolateral function regardless.

Our models of mechanical advantage do not account for any cranial kinesis. This is a necessary simplifcation. The nature of cranial kinesis in extant birds, and the role that it plays in feeding, is poorly understood. Cranial kinesis will increase gape, but we don’t yet know how/if it affects jaw closing force and speed (moreover, given the variation in quadrate and hinge morphology present in extant birds, this is also something that is likely to be highly diverse). We have therefore modelled the extant birds’ jaw closing systems as having one, akinetic out lever (the jaw joint to the bite point), to match the situation in our fossil taxa. This is a common simplification that has been used previously with success (Corbin, Lowenberger et al. 2015, Olsen 2017). However, we acknowledge that this simplification may introduce some error. Unfortunately, until the mechanics of cranial kinesis – and the variation in the anatomy and performance of kinetic structures in extant birds – are better understood, we cannot determine exactly what that error looks like. We therefore have greater confidence in the inter-species comparability this conservative, akinetic approach (in other words, we may not be making assumptions that are 100% accurate, but we are at least making the same assumption across all taxa, so it should be comparable in its error). We will add a section in the Mechanical Advantage and Functional Indices discussion calling for further research into the mechanics of cranial kinesis so future mechanical advantage work in birds can take this matter into account.

• Reviewer 3, on skull reconstruction: This issue is partly addressed in the Bohaiornithid Skull Reconstruction section, though we agree that adding more mentions of it in the MA and FEA Discussion sections and the Bohaiornithid Ecology and Evolution sections will benefit the manuscript. Most notably Shenqiornis and Sulcavis have similar ecological interpretations, but much of the Shenqiornis skull reconstruction uses Sulcavis bones. *Longusunguis* is the only other taxon which takes more than two bones from a different taxon, and in this case all but the quadrate are not used in any quanitative measurements. We have ensured that the skull reconstructions presented in Figure 2 show what portions of the skull come from what specimen so that as new material is discovered and phylogenetic relationships are updated it will be clear to future readers which parts of reconstructions will need to be updated.

• Reviewer 3, on data availability: All data including FEA models and raw measurement data are included in the same repository as the scripts, which we will make clear in the manuscript. Good catch on the data link being dead, we will publish it now.

As a final note, it was brought to our attention by another colleague that the original manuscript’s ancestral state reconstrction lacked an outgroup. An updated reconstruction using *Sapeornis* as an outgroup will be included in the revised manuscript. The addition of the outgroup does not change any conclusions of the manuscript.

We once again thank our reviewers for their valuable feedback and will submit a revised version of this manuscript for publication shortly. Please let us know if you have any additional comments after reading our response that we can take onboard in our revision.

References

Artuso, C., C. S. Houston, D. G. Smith and C. Rohner (2020). Great Horned Owl (*Bubo virginianus*), version 1.0. Birds of the World. A. F. Poole. Ithaca, NY, USA, Cornell Lab of Ornithology.

Atterholt, J., A. W. Poust, G. M. Erickson and J. K. O'Connor (2021). "Intraskeletal osteohistovariability reveals complex growth strategies in a Late Cretaceous enantiornithine." Frontiers in Earth Science 9: 640220.

Corbin, C. E., L. K. Lowenberger and B. L. Gray (2015). "Linkage and trade‐off in trophic morphology and behavioural performance of birds." Functional ecology 29(6): 808-815.

Hone, D. W. E., A. A. Farke and M. J. Wedel (2016). "Ontogeny and the fossil record: what, if anything, is an adult dinosaur?" Biology letters 12(2): 20150947.

Hu, H. and J. K. O'Connor (2017). "First species of Enantiornithes from Sihedang elucidates skeletal development in Early Cretaceous enantiornithines." Journal of Systematic Palaeontology 15(11): 909-926.

Miller, C. V., M. Pittman, X. Wang, X. Zheng and J. A. Bright (2023). "Quantitative investigation of Mesozoic toothed birds (Pengornithidae) diet reveals earliest evidence of macrocarnivory in birds." iScience 26(3): 106211.

Olsen, A. M. (2017). "Feeding ecology is the primary driver of beak shape diversification in waterfowl." Functional Ecology 31(10): 1985-1995.

Schnell, J. H. (2020). Common Black Hawk (*Buteogallus anthracinus*), version 1.0. Birds of the World. A. F. Poole and F. B. Gill. Ithaca, NY, USA, Cornell Lab of Ornithology.